# Accelerating volcanic ash data assimilation using a mask-state algorithm based on ensemble Kalman filter: a case study with the LOTOS-EUROS model (version 1.10)

Guangliang Fu[1], Hai Xiang Lin[1], Arnold Heemink[1], Sha Lu[1], Arjo Segers[2], Nils van Velzen[1,3], Tongchao Lu[4], and Shiming Xu[5]

[1]Delft University of Technology, Delft Institute of Applied Mathematics, Mekelweg 4, 2628 CD Delft, The Netherlands.
[2]TNO, Department of Climate, Air and Sustainability, P.O. Box 80015, 3508 TA Utrecht, The Netherlands.
[3]VORtech, P.O. Box 260, 2600 AG Delft, The Netherlands.
[4]School of Mathematics, Shandong University, Jinan, Shandong, China.
[5]Department of Earth System Science, Tsinghua University, Beijing, China

*Correspondence to:* Guangliang Fu (G.Fu@tudelft.nl)

**Abstract.** In this study, we investigate a strategy to accelerate the data assimilation (DA) algorithm. Based on evaluations of the computational time, the analysis step of the assimilation turns out to be the most expensive part. After a study on the characteristics of the ensemble ash state, we propose a mask-state algorithm which records the sparsity information of the full ensemble state matrix and transforms the full matrix into a relatively small one. This will reduce the computational cost in the analysis step. Experimental results show the mask-state algorithm significantly speeds up the analysis step. Subsequently, the total amount of computing time for volcanic ash DA is reduced to an acceptable level. The mask-state algorithm is generic and thus can be embedded in any ensemble-based DA framework. Moreover, ensemble-based DA with the mask-state algorithm is promising and flexible, because it implements exactly the standard DA without any approximation and it realizes the satisfying performance without any change of the full model.

## 1 Introduction

Volcanic ash erupted into atmospheres can lead to severe influences on aviation society (Gudmundsson et al., 2012). Turbine engines of airplanes are extremely threatened by the ash's ingestion (Casadevall, 1994). Thus, accurate real-time aviation advices are highly required during an explosive volcanic ash eruption (Eliasson et al., 2011). Recently, ensemble-based DA (Evensen, 2003) has been evaluated very useful to improve volcanic ash forecasts and regional aviation advices (Fu et al., 2016). It corrects volcanic ash concentrations by continuously assimilating observations. In (Fu et al., 2016), real aircraft in situ measurements were assimilated using the ensemble Kalman filter (EnKF), which is the most known and popular ensemble-based DA method. Based on the validation with independent data, ensemble-based DA was concluded powerful for improving the forecast accuracy.

However, to make the methodology efficient also in an operational (real-time) sense, the computational efforts must be acceptable. For volcanic ash DA problems, so far, no studies on the computational aspects have been reported in the literature.

Actually, when large amounts of volcanic ash erupted into atmospheres, the computational speed of volcanic ash forecasts is just as important as the forecast accuracy (Zehner, 2010). For example, due to the lack of a fast and accurate forecast system, the sudden eruption of the Eyjafjallajökull volcano in Iceland from 14 April to 23 May 2010, had caused an unprecedented closure of the European and North Atlantic airspace resulting in a huge global economic loss of 5 billion US dollars (Oxford-Economics, 2010). Since then, research on fast and accurate volcanic ash forecasts have gained much attention, because it is needed to provide timely and accurate aviation advices for frequently operated commercial airplanes. It was shown the accuracy of volcanic ash transport can be significantly improved by the DA system in (Fu et al., 2016). Therefore, it is urgent to also consider the computational aspect, i.e., improving the computational speed of the volcanic ash DA system as fast as possible. This is the main focus of this study.

Due to the computational complexity of ensemble-based algorithms and the large scale of dynamical applications, applying these methods usually introduces a large computational cost. This has been reported from literature on different applications. For example, for operational weather forecasting with ensemble-based DA, Houtekamer et al. (2014) reported computational challenges at Canadian Meteorological Center with an operational EnKF featuring 192 ensemble members, using a large $600 \times 300$ global horizontal grid and 74 vertical levels. That an initialization requirement of over $7 \times 10^{10}$ values to specify each ensemble, results in large computational efforts on the initialization and forecast steps in weather forecasting. For oil reservoir history-matching (Tavakoli et al., 2013), the reservoir simulation model usually has a large number of state variables, thus the forecasts of an ensemble of simulation models are often time-consuming. Besides, when time-lapse seismic or dense reservoir data is available, the analysis step of assimilating these large observations becomes very time-consuming (Khairullah et al., 2013). Large computational requirements of ensemble-based DA have also been reported in ocean circulation models (Keppenne, 2000; Keppenne and Rienecker, 2002), tropospheric chemistry assimilation (Miyazaki et al., 2015), and many other applications.

To accelerate an ensemble-based DA system, the ensemble forecast step can be first parallelized because the propagation of different ensemble members is independent. Thus if a computer with a sufficiently large number of parallel processors is available, all the ensemble members can be simultaneously integrated. In the analysis stage, to calculate the Kalman gain and the ensemble error covariance matrix, all ensemble states must be combined together. In weather forecasting and oceanography sciences, Keppenne (2000); Keppenne and Rienecker (2002); Houtekamer and Mitchell (2001) have reported using parallelization approaches to accelerate the expensive analysis stage. In reservoir history matching, a three-level parallelization has been proposed by Tavakoli et al. (2013); Khairullah et al. (2013) in recent years, to significantly reduce computational efforts of both forecast and analysis steps due to massive dense observations and large simulation models. The first parallelization level is to separately perform the ensemble simulations on different processors during the forecast step. This approach is usually quite efficient when a large ensemble size is used. However, the scale or model size of one reservoir simulation is constrained by the memory of a single processor. Thus, the second parallelization level is to perform one ensemble member simulation using a parallel reservoir model. These two levels do not deal with the analysis step, which collects all ensemble members to do computations usually on a single processor. Therefore, a third level of parallelization was implemented by Tavakoli et al. (2013); Khairullah et al. (2013) through parallelizing matrix-vector multiplications in the analysis steps. Furthermore,

some other approaches on accelerating ensemble-based DA systems, have also been reported, such as GPU-based acceleration (Quinn and Abarbanel, 2011) in Numerical Weather Prediction (NWP), domain decomposition in atmospheric chemistry assimilation (Segers, 2002; Miyazaki et al., 2015). The observations used in an DA system can be also optimized with some preprocessing procedures, as reported by Houtekamer et al. (2014).

Although for other applications, there were many efforts in dealing with large computational requirements in an ensemble-based DA system, most of them cannot be directly used to accelerate volcanic ash DA. This is because the acceleration algorithms are strongly dependent on specific problems, such as model complexity (high or low resolution), observation type (dense or sparse), primary requirement (accuracy or speed). These factors determine, for a specific application, which part is the most time-consuming, and which part is intrinsically sequential. Thus, no unified approach for efficient acceleration of all the applications can be found. Although the successful approaches in other applications cannot be directly employed in volcanic ash forecasts, their success do stress the importance of designing a proper approach based on the computational analysis of a specific DA system. Therefore, the computational cost of our volcanic ash DA system will be first analyzed. Then, based on the computational analysis, we will investigate a strategy to accelerate the ensemble-based DA system for volcanic ash forecasts.

This paper is organized as follows. Sect. 2 introduces the methodology of volcanic ash DA. Sect. 3 analyzes the computational cost of the conventional volcanic ash DA system. In Sect. 4, the mask-state algorithm (MS) is developed for acceleration. The comparison between MS and standard sparse matrix methods are presented in Sect. 5. The discussions on MS is in Sect. 6. Finally, the last section summarizes the concluding remarks of our research.

## 2 Methodology of the volcanic ash DA system

In this study, the ensemble Kalman filter (EnKF) (Evensen, 2003) is employed to perform ensemble-based DA. EnKF is typically a sequential Monte Carlo method, according to the uncertain state estimate with $N$ ensemble members, $\boldsymbol{\xi}_1, \boldsymbol{\xi}_2, \cdots, \boldsymbol{\xi}_N$. Each member is assumed as one sample in the distribution of the true state. It has been proposed that for operational applications, the ensemble size can be limited to $10\text{-}100$ for cost effectiveness (Nerger and Hiller, 2013; Barbu et al., 2009). Thus, in this study, an ensemble size of 100 is used due to the high-accuracy requirement of the volcanic ash forecasts to aviation advices as mentioned in Sect. 1.

To simulate a volcanic ash plume, an atmospheric transport model is needed. In this paper, the LOTOS-EUROS (abbreviation of LOng Term Ozone Simulation - EURopean Operational Smog) model is used (Schaap et al., 2008) with model version 1.10 (http://www.lotos-euros.nl/). The LOTOS-EUROS model (Schaap et al., 2008) is an operational model focusing on nitrogen oxides, ozone, particular matter, volcanic ash. The model configurations for volcanic ash were discussed in details by Fu et al. (2016). For volcanic ash simulation, the model is configured with a state vector of size $180 \times 200 \times 18 \times 6$ (the dimensions correspond to longitude, latitude, vertical level, ash species), the size of model state is thus calculated as $\sim 10^6$.

The experiment in this study starts at $t_0$ (09:00 UTC, 18 May, 2010 for this study) by considering an initial condition from previous LOTOS-EUROS conventional model run (see Fig. 1(a)). In the second step (the forecast step) the model propagates

the ensemble members from the time $t_{k-1}$ to $t_k$ ($k > 0$, the time step is 10 minutes):

$$\boldsymbol{\xi}_j^f(k) = M_{k-1}(\boldsymbol{\xi}_j^a(k-1)). \tag{1}$$

The operator $M_{k-1}$ describes the time evolution of the state which contains the ash concentrations in all model grid-boxes. The state at the time $t_k$ has a distribution with the mean $\boldsymbol{x}^f$ and the forecast error covariance matrix $\mathbf{P}^f$ given by:

$$\boldsymbol{x}^f(k) = [\sum_{j=1}^{N} \boldsymbol{\xi}_j^f(k)]/N, \tag{2}$$

$$\mathbf{L}^f(k) = [\boldsymbol{\xi}_1^f(k) - \boldsymbol{x}^f(k), \cdots, \boldsymbol{\xi}_N^f(k) - \boldsymbol{x}^f(k)], \tag{3}$$

$$\mathbf{P}^f(k) = [\mathbf{L}^f(k)\mathbf{L}^f(k)^T]/(N-1), \tag{4}$$

where $\mathbf{L}^f$ represents the ensemble perturbation matrix. In this study, the forecast step is performed in parallel because of the natural/common parallelism of the independent ensemble propagation, which is a trivial approach when employing ensemble-based DA (Liang et al., 2009; Tavakoli et al., 2013; Khairullah et al., 2013).

When the model propagates to 09:40 UTC, 18 May, 2010, the volcanic ash state gets sequentially analyzed by the DA process through combining real aircraft in situ measurements of $PM_{10}$ and $PM_{2.5}$ concentrations until 11:10 UTC. The measurement route and values are demonstrated in Figs. 1(b, c) and the details are described in (Weber et al., 2012; Fu et al., 2016). The observational network at time $t_k$ is defined by the operator $H_k$ which maps the state vector $\boldsymbol{x}$ to the observational vector $\boldsymbol{y}$ by

$$\boldsymbol{y}(k) = H_k(\boldsymbol{x}(k)) + \boldsymbol{v}(k), \tag{5}$$

where $\boldsymbol{y}$ contains the aircraft measurements and $\boldsymbol{v}$ represents the observational error. $H_k$ selects the grid cell in $\boldsymbol{x}(k)$ that corresponds to the locations of the observation. When measurements are available, the ensemble members are updated in the analysis step using

$$\mathbf{K}(k) = \mathbf{P}^f(k)\mathbf{H}(k)^T[\mathbf{H}(k)\mathbf{P}^f(k)\mathbf{H}(k)^T + \mathbf{R}]^{-1}, \tag{6}$$

$$\boldsymbol{\xi}_j^a(k) = \boldsymbol{\xi}_j^f(k) + \mathbf{K}(k)[\boldsymbol{y}(k) - \mathbf{H}(k)\boldsymbol{\xi}_j^f(k) + \boldsymbol{v}_j(k)], \tag{7}$$

where $\mathbf{K}$ represents the Kalman gain, $\mathbf{H}$ is the observational matrix formed by the observational operator $H$, $\mathbf{R}$ represents the measurement error covariance matrix and $\boldsymbol{v}_j$ represents the realization out of the observation error distribution $\boldsymbol{v}$. After the continuous assimilation ending at 11:10 UTC, the forecast at 12:00 UTC is illustrated in Fig. 1(d), for which the forecast accuracy has been carefully evaluated as significantly improved compared to the case without DA (Fu et al., 2016).

The EnKF with above setups is abbreviated as "conventional EnKF" and used in this study for the computational evaluation. Note that in the study we don't use covariance localization as proposed by Hamill et al. (2001) for reducing spurious covariances. This is because although localization is possible, the ideal case is not to use it in order to have the correct covariances in a large (converged) ensemble. It is crucial for localization that when unphysical (spurious) covariances are eliminated, physical (correct) covariances can be well maintained (Petrie and Dance, 2010). If the "filtering length scale" for localization is too long (i.e., all the dynamical covariances are allowed), many of the spurious covariances may not be eliminated. If the length is

too short, important physical dynamical covariances then may be lost together with the spurious ones. Therefore, essentially deciding an accurate localization is a challenging subject (Riishojgaard, 1998; Kalnay et al., 2012) especially for accuracy-demanding applications. Therefore, in this study we choose the ensemble size of 100 to guarantee the accuracy and avoid large spurious covariances.

## 3  Computational analysis for volcanic ash DA

### 3.1  Computational analysis of the total runtime

Ensemble-based DA is a useful approach to improve the forecast accuracy of volcanic ash transport. However, if it is time-consuming, it cannot be taken as efficient due to the high requirement on speed for volcanic ash DA (see Sect. 1). Based on this consideration, we need to analyze the computational cost of a conventional volcanic ash DA system.

As introduced in Sect. 2, the total execution time of conventional EnKF comprises four parts, i.e., initialization, forecast, analysis and other computational cost. The initialization time includes reading meteorological data, initializing model geographical and grid configurations, reading emission information, initializing stochastic observer for reading and transforming observations to the model grid, initializing all the ensemble states and ensemble mean, and so on. The forecast time is obtained from Eq. (1), while the analysis time corresponds to the computational sum from Eq. (2) to (7). The other computational time includes script compiling, setting environment variables, starting and finalizing DA algorithms, etc.

The evaluation result of the conventional EnKF is shown in Table 1 (the middle column). It can be seen that the total computational time (4.36 h) is relatively large compared to the simulation window (3.0 h, i.e., from 9:00 - 12:00 UTC, 18 May, 2010), which is too much in an operational sense. Therefore, in this study, we aim to accelerate the computation to within an acceptable runtime (i.e., requires less runtime than the time period of the DA application).

It can be also observed from Table 1 that the main contribution to the total execution time is the analysis step. Compared to the initialization and forecast time, the analysis stage takes 72 % of the total runtime. Due to the expensive analysis step, although some approaches (such as MPI-parallel I/O (Filgueira et al., 2014), domain decomposition (Segers, 2002)) can potentially accelerate the initialization and forecast step, the effect to the final acceleration of the total computational cost is little. Therefore, to get acceptable computational time, the cost reduction in the analysis step is the target. One may wonder that since the amount of observations is small, why does analysis takes so much time? The large state vector seems to be left responsible for the problem. To know the exact reason, the detailed computational cost of the analysis step must be evaluated.

### 3.2  Cost estimation of all analysis procedures

We start with the formulations of the analysis step. The analysis step is represented by Eq. (7), which can be written in a full matrix format with Eq. (8),

$$\mathbf{A}_{n \times N}^a = \mathbf{A}_{n \times N}^f + \mathbf{K}_{n \times m}(\mathbf{Y}_{m \times N} - \mathbf{H}_{m \times n}\mathbf{A}_{n \times N}^f), \tag{8}$$

where the subscripts represent the matrix's dimensions. $\mathbf{A}^f$ and $\mathbf{A}^a$ represent the forecasted and analyzed ensemble state matrix, and are respectively built up from $\boldsymbol{\xi}^f$ and $\boldsymbol{\xi}^a$ with $N$ ensembles. The measurement ensemble matrix $\mathbf{Y}$ is formed by an ensemble of $\boldsymbol{y} + \boldsymbol{v}$ (see Eq. (7)). $\mathbf{H}$ is the observational matrix, which is used to select state variables (at measurement locations) in the full ensemble state matrix corresponding to the measurement ensemble matrix $\mathbf{Y}$. $n$ is the number of model

state variables in a three-dimensional (3-D) domain, i.e., $\sim 10^6$ in this study (see Sect. 2). $m$ is the amount of measurements at one assimilation time, which depends on the measurement type. For aircraft in situ measurements used in this study (see Fig. 1(c)), two measurements are made at each time by one research flight, so that $m$ is 2 here. $N$ is the ensemble size and is taken as 100 in this study. As described in Eq. (3), the ensemble perturbation matrix $\mathbf{L}^f$ in EnKF can be re-written as

$$\mathbf{L}_{n \times N}^f = \mathbf{A}_{n \times N}^f - \bar{\mathbf{A}}_{n \times N}^f = \mathbf{A}_{n \times N}^f (\mathbf{I}_{N \times N} - \frac{1}{N}\mathbf{1}_{N \times N}) = \mathbf{A}_{n \times N}^f \mathbf{B}_{N \times N}, \tag{9}$$

where $\mathbf{I}$ is an $N \times N$ unit matrix and $\mathbf{1}$ is an $N \times N$ matrix with all elements equal to 1. Thus, $\mathbf{L}^f = \mathbf{A}^f \mathbf{B}$ where $\mathbf{B}_{N \times N}$ is introduced to represent $(\mathbf{I}_{N \times N} - \frac{1}{N}\mathbf{1}_{N \times N})$. So that, $\mathbf{HL}^f = \mathbf{O}^f \mathbf{B}$, where $\mathbf{O}_{m \times N}^f$ is used to represent $(\mathbf{HA}^f)$. Here we explicitly express $\mathbf{L}^f$ and $\mathbf{HL}^f$ in the form of $\mathbf{A}^f$ and $\mathbf{O}^f$, respectively. This is because in our volcanic ash DA system, $\mathbf{A}^f$ and $\mathbf{O}^f$ are two of the three inputs (another one is the measurement ensemble matrix $\mathbf{Y}$ for the analysis step. These are the three inputs used for actual computations in the analysis step. As shown in Fig. 2(a), $\mathbf{A}^f$ is obtained from the forecast step,

$\mathbf{O}^f$ and $\mathbf{Y}$ are acquired from our stochastic observer module (see Fig. 2(a)) which is used for a volcanic ash transport model to integrate geophysical measurements. With the input $\mathbf{Y}$, the measurement error covariance $\mathbf{R}$, as introduced in Eq. (6), can be then computed with

$$\mathbf{R}_{m \times m} = \frac{1}{N-1}(\mathbf{Y}_{m \times N} - \bar{\mathbf{Y}}_{m \times N})(\mathbf{Y}_{m \times N} - \bar{\mathbf{Y}}_{m \times N})' = \frac{1}{N-1}(\mathbf{YB})(\mathbf{YB})'. \tag{10}$$

Based on previous definitions and Eqs. (2) to (7), the analysis step can be reformulated as followings,

$$\begin{aligned}
\mathbf{A}_{n \times N}^a &= \mathbf{A}^f + \mathbf{K}(\mathbf{Y} - \mathbf{HA}^f) \\
&= \mathbf{A}^f + \mathbf{P}^f \mathbf{H}'(\mathbf{HP}^f \mathbf{H}' + \mathbf{R})^{-1}(\mathbf{Y} - \mathbf{HA}^f) \\
&= \mathbf{A}^f + \frac{1}{N-1}\mathbf{L}^f(\mathbf{HL}^f)'[\frac{1}{N-1}(\mathbf{HL}^f)(\mathbf{HL}^f)' + \frac{1}{N-1}(\mathbf{YB})(\mathbf{YB})']^{-1}(\mathbf{Y} - \mathbf{HA}^f) \\
&= \mathbf{A}^f + \mathbf{A}^f \mathbf{B}(\mathbf{O}^f \mathbf{B})'[(\mathbf{O}^f \mathbf{B})(\mathbf{O}^f \mathbf{B})' + (\mathbf{YB})(\mathbf{YB})']^{-1}(\mathbf{Y} - \mathbf{O}^f) \\
&= \mathbf{A}^f \{\mathbf{I} + \mathbf{B}(\mathbf{O}^f \mathbf{B})'[(\mathbf{O}^f \mathbf{B})(\mathbf{O}^f \mathbf{B})' + (\mathbf{YB})(\mathbf{YB})']^{-1}(\mathbf{Y} - \mathbf{O}^f)\} \\
&= \mathbf{A}_{n \times N}^f \mathbf{X}_{N \times N} \quad ,
\end{aligned} \tag{11}$$

where

$$\mathbf{X}_{N \times N} = \{\mathbf{I} + \mathbf{B}(\mathbf{O}^f \mathbf{B})'[(\mathbf{O}^f \mathbf{B})(\mathbf{O}^f \mathbf{B})' + (\mathbf{YB})(\mathbf{YB})']^{-1}(\mathbf{Y} - \mathbf{O}^f)\}. \tag{12}$$

Eq. (11) shows how the analysis step is performed in a volcanic ash DA system. In order to accelerate the analysis step, the most time-consuming part must be reduced. Fig. 2(b) shows estimations of the computational cost for each procedure in the analysis

step. Considering that the state number $n$ ($\sim 10^6$) is significantly larger than the measurement number $m$ ($m$=2 here) and the

ensemble size $N$ ($N$=100), thus the most time-consuming procedure in the analysis step is the last one, that is $\mathbf{A}^a = \mathbf{A}^f\,\mathbf{X}$ with computational cost of O($nN^2$). Therefore, in our volcanic ash DA system, this part is the most time-consuming part in the analysis step. Note that the procedure $[(\mathbf{O}^f\mathbf{B})(\mathbf{O}^f\mathbf{B})' + (\mathbf{YB})(\mathbf{YB})']^{-1}$ for Singular Value Decomposition (SVD) in our study is not time-consuming, which is different from applications of reservoir history matching (Tavakoli et al., 2013; Khairullah et al., 2013). This is because the SVD procedure costs O($m^3$), and due to the measurement size in the order of the size of the state in those cases, SVD procedure thus requires a huge computational cost for reservoir DA.

## 4 The mask-state algorithm (MS) for acceleration of the analysis step

### 4.1 Characteristic of ensemble state matrix $\mathbf{A}^f$

Analysis in the previous section shows that $\mathbf{A}^a = \mathbf{A}^f\,\mathbf{X}$ is most expensive in the analysis step. Each column of $\mathbf{A}^f$ is constructed from a forecasted ensemble state, thus the dimension of $\mathbf{A}^f$ is $n \times N$. In each column, the element values correspond to volcanic ash concentrations in a 3-D domain. Fig. 3 shows the coverage of all ensemble forecast states at a selected time 10:00 UTC 18 May, 2010, without loss of generality. A common phenomenon can be observed, that is only a part of the 3-D domain are filled with volcanic ash. The ash clouds only concentrate in a plume which is transported over time. This is because volcanic eruption is a fast and strong process. The advection dominates the transport, and the volcanic ash plume is transported with the wind. This is a particular characteristic for volcanic ash transport, in contrast to other atmospheric related applications such as ozone (Curier et al., 2012), $SO_2$ (Barbu et al., 2009), $CO_2$ (Chatterjee et al., 2012). For those applications, the concentrations are everywhere in the domain, the emission sources are also everywhere, and observations are available throughout the domain too (especially for satellite data). Whereas for application of volcanic ash transport, the source emission is only at the volcano, thus usually only a limited domain is polluted by ash. As shown in Fig. 3, in the 3-D domain with grid size of $3.888 \times 10^6$, the number of grids in the area with volcanic ash is counted as $1.528 \times 10^6$, whereas the number of no-ash grids is $2.36 \times 10^6$. Note that shown in the figure are accumulated ash coverages of all ensemble states, thus in the no-ash grids, there are no ash for all the ensemble states. Thus a very large number of rows in $\mathbf{A}^f$ are zero corresponding to the no-ash grids. These zero rows in $\mathbf{A}^f$ have no contributions to $\mathbf{A}^a = \mathbf{A}^f\,\mathbf{X}$, because a zero row in $\mathbf{A}^f$ always results in a zero row in $\mathbf{A}^a$. Therefore, for the case of Fig. 3, $\frac{2}{3}$ of the computations are redundant and can be avoided. To realize this, one may think to limit the domain for the entire assimilation steps, then the number of zero rows certainly would be largely reduced. This is actually incorrect, because these zero rows are changing along with the transport of ash clouds, and not constant at each analysis step. So the full domain must be considered and it should be adaptive (choose different zero rows according to different $\mathbf{A}^f$ at different analysis time).

### 4.2 Derivation of the mask-state algorithm (MS)

Here we introduce item $n_{noash}$ to represent the number of zero rows in the ensemble state matrix $\mathbf{A}^f$, and use $n_{ash}$ to represent the number of other rows (also $n_{ash}$ represents the grid size of ash plume). When computing $\mathbf{A}^a = \mathbf{A}^f\,\mathbf{X}$, to avoid

all the computations related to $n_{noash}$ rows with zero elements, the index of other $n_{ash}$ rows must be first decided. This index is meant to reduce the dimensions of $\mathbf{A}^f$. After getting a $\mathbf{A}^a$ with a dimension of $n_{ash} \times N$, the index will be used again to reconstruct the full matrix $\mathbf{A}^a$ with the dimension of $n \times N$. Based on this idea, we propose a mask-state algorithm (MS) which deals with the time-consuming analysis update. MS includes five steps:

(i) **Compute ensemble mean state $\bar{\mathbf{A}}^f$**: The mean state $\bar{\mathbf{A}}^f_{n \times 1}$ can be easily computed by averaging $\mathbf{A}^f_{n \times N}$ along $N$ columns. Due to all elements in $\mathbf{A}^f_{n \times N}$ corresponding to ash concentrations, thus all elements in $\mathbf{A}^f_{n \times N}$ are larger than or equal to zero. The index of non-zero rows in $\bar{\mathbf{A}}^f_{n \times 1}$ is thus equivalent to that in $\mathbf{A}^f_{n \times N}$. The computational cost for this step is O($nN$).

    (ii) **Construct mask array $\mathbf{z}$**: Based on previously obtained $\bar{\mathbf{A}}^f_{n \times 1}$, we search the non-zero elements of $\bar{\mathbf{A}}^f_{n \times 1}$ and record

the index into a mask array $\mathbf{z}_{n_{ash} \times 1}$. With this strategy, we don't need to search the full matrix $\mathbf{A}^f_{n \times N}$ and build an index matrix for storage. This is a benefit for saving memory. The computational cost for this step is O($n$).

    (iii) **Construct masked ensemble state matrix $\tilde{\mathbf{A}}^f$**: Using the mask array $\mathbf{z}_{n_{ash} \times 1}$ obtained from step (ii), $\tilde{\mathbf{A}}^f_{n_{ash} \times N}$ can be constructed column by column according to Eq. (13), and the computational cost (overhead) for this step is O($n_{ash} N$).

$$\tilde{\mathbf{A}}^f(1 : n_{ash}, 1 : N) = \mathbf{A}^f(\mathbf{z}(1 : n_{ash}), 1 : N), \tag{13}$$

(iv) **Compute $\tilde{\mathbf{A}}^a$ by multiplying $\tilde{\mathbf{A}}^f$ and $\mathbf{X}$**: Perform matrix computation $\tilde{\mathbf{A}}^a_{n_{ash} \times N} = \tilde{\mathbf{A}}^f_{n_{ash} \times N} \mathbf{X}_{N \times N}$. This step is similar to $\mathbf{A}^a = \mathbf{A}^f \mathbf{X}$, as described in Sect. 3.2, but the computational cost now becomes O($n_{ash} N^2$) instead of O($n N^2$).

    (v) **Construct analyzed ensemble state matrix $\mathbf{A}^a$**: With the computed $\tilde{\mathbf{A}}^a$ from step (iv) and the mask array $\mathbf{z}$ from step (ii), the final analyzed ensemble state matrix $\mathbf{A}^a_{n \times N}$ can be constructed based on Eq. (14). The computational cost (overhead) for this step is O($nN$).

$$\mathbf{A}^a(\mathbf{z}(1 : n_{ash}), 1 : N) = \tilde{\mathbf{A}}^a(1 : n_{ash}, 1 : N), \tag{14}$$

According to the derivations of MS, the computational cost related to zero rows are avoided. Here the "zero rows" doesn't equal to "zero elements". The former corresponds to the regions where there are no ash for all the ensemble members, while the latter also counts the no-ash regions specifically for some ensembles. Certainly the consideration of all "zero elements" can include all the sparsity information of the ensemble state matrix, but extra computations and memories must be spent on

searching the full matrix $\mathbf{A}^f_{n \times N}$ with a computational cost of O($nN$) and storing a mask state matrix with dimensions of $n \times N$. This is expensive compared to construct the mask array in the procedure (ii). Actually, after a careful check on the volcanic ash ensemble plumes, there is no "bad" ensemble which is really different from others. Although the concentration level in ensemble members are distinct, the main direction and the occurrence to the grid cells are more or less same. This means, the "zero rows" actually more or less equals to "zero elements", but much faster than the way with "zero elements",

which confirms the suitability and advantage of procedure (ii). Probably when there are big meteorological uncertainties, the

"zero elements" will be much larger than "zero rows". In this case, how to make use of the sparsity information in the ensemble state matrix, will be considered in future.

Based on procedures of MS, the computational cost of $\mathbf{A}^a = \mathbf{A}^f \mathbf{X}$ can be reduced. However, without a careful evaluation, we cannot conclude MS is fast, because the algorithm also employs other procedures. If these procedures (i)(ii) (iii)(v) are
much cheaper than the main procedure (iv), MS can definitely speed up the analysis step, and vice versa. Now we analyze MS's computational cost, which can be summed as $O(n\,N)$+ $O(n)$+ $O(n_{ash}\,N)$+ $O(n_{ash}\,N^2)$+ $O(n\,N)$, i.e., $O(n\,N+n_{ash}\,N^2)$. Thus, the computational overhead involved to transform the full matrix to a small one (i.e., $O(n_{ash}\,N)$ for procedure (iii)) has little effect in the total computation cost of MS (i.e., $O(n\,N+n_{ash}\,N^2)$). However, the computational overhead of transforming the small matrix to the full one (i.e., $O(n\,N)$ for procedure (v)) does contribute a part, which cannot be ignored, to the total
MS's computational cost. The computational cost without MS is $O(n\,N^2)$.

The comparison between both cost (with and without MS, i.e., $O(n\,N+n_{ash}\,N^2)$ and $O(n\,N)$) indicates when the number of non-zero rows ($n_{ash}$, i.e., the number of grids with ash) of the forecasted ensemble state matrix satisfies $n_{ash} < \frac{N-1}{N}n$, then MS can accelerate $\mathbf{A}^a = \mathbf{A}^f \mathbf{X}$. Here, $O(n\,N+n_{ash}\,N^2)$ and $O(n\,N)$ are of the same order when $n_{ash} < \frac{N-1}{N}n$. The larger the difference between $n_{ash}$ and $\frac{N-1}{N}n$, the better the speedup can be achieved. According to this analysis, and the
characteristic (e.g., $\frac{n_{ash}}{n}$ approximately equals to $\frac{1}{3}$ in this case) of volcanic ash transport as described in Sect. 4.1, the relation is certainly satisfied and is actually $n_{ash} \ll \frac{N-1}{N}n$ (significantly smaller) for our study. Therefore, for our volcanic ash DA system, with MS, the computational cost for the time-consuming part $\mathbf{A}^a = \mathbf{A}^f \mathbf{X}$ is $O(n_{ash}\,N^2)$, which is much reduced compared to $O(n\,N^2)$ with conventional computations.

The relation $n_{ash} < \frac{N-1}{N}n$ indicates whether we would have speedup by the MS method, actually it can be extended to Eq.
20  (15),

$$S_{\mathrm{ms}} = \frac{O(n\,N^2)}{O(n\,N+n_{ash}\,N^2)} = O(\frac{n}{n_{ash}}), \tag{15}$$

which explicitly specifies the expected amount of speedup ($S_{\mathrm{ms}}$) of $\mathbf{A}^a = \mathbf{A}^f \mathbf{X}$ by the MS algorithm. In this case study, $N$ is taken at 100 and $\frac{n_{ash}}{n} \approx \frac{1}{3}$, so $S_{\mathrm{ms}}$ is approximately 3.0.

According to Amdahl's law (Amdahl, 1967), the total computational speedup ($S_{\mathrm{total}}$) by MS can be predicted by Eq. (16),

$$S_{\mathrm{total}} = \frac{1}{(1-p_{\mathrm{ms}})+\frac{p_{\mathrm{ms}}}{S_{\mathrm{ms}}}}, \tag{16}$$

where $p_{\mathrm{ms}}$ is the proportion of the computational cost of $\mathbf{A}^a = \mathbf{A}^f \mathbf{X}$ in the overall DA computations. It has been evaluated that the computational cost of $\mathbf{A}^a = \mathbf{A}^f \mathbf{X}$ dominates the analysis step (see Fig. 2(b)), thus the proportion of the computational cost of $\mathbf{A}^a = \mathbf{A}^f \mathbf{X}$ approximates the proportion of the analysis step in the total DA computations (i.e., $p_{\mathrm{ms}} \approx 72\,\%$ in this case, as described in Sect. 3.1). Therefore, based on Eq. (16), the maximum ("ideal") computational speedup can be predicted
to be $\frac{1}{1-p_{\mathrm{ms}}}$ (i.e., $\approx 3.57$ for this case study) when $S_{\mathrm{ms}}$ approximates infinity. However, this is not the actual speedup because $S_{\mathrm{ms}}$ is in fact specified by Eq. (15). Based on discussions above, $S_{\mathrm{total}}$ can be therefore estimated by Eq. (15) at $\approx 2.0$ in this case.

## 4.3 Experimental results

Analysis of the algorithmic complexity of MS shows that MS is an efficient approach to reduce the computational cost of the time-consuming $\mathbf{A}^a = \mathbf{A}^f \mathbf{X}$. Now MS will be applied in the real volcanic ash DA system, to investigate whether in practice it can well speed up the analysis step. We perform MS in the conventional EnKF, which means initialization, forecast steps are all computed as the conventional EnKF. The only difference between MS-EnKF and conventional EnKF is that in the former MS is employed for analysis step, and in the latter is standard analysis step. The result and related specifications are shown in Table 1. As introduced in Sect. 2, the forecast step has been configured with the conventional parallelization, thus $N+2$ (102 here) cores are actually used (one core for the DA algorithm, the other $N+1$ cores for the parallel forecast of $N$ ensemble members and one ensemble mean). It can be seen from Table 1 that MS indeed largely accelerates the analysis step (as expected, by a factor of about 3.0 for this study) which confirms the theoretical cost evaluation. The detailed experimental time for each step of MS is shown in Table 2. As expected, the dense-dense matrix multiplication in step (iv) takes the largest part (i.e., 0.8474 h for this case study) of the total computational time (0.87 h) of MS. However, step (iv) has been a big improvement compared to the case without MS (3.14 h, see Table 1), which is because the computational time for other steps (e.g., steps (i - iii) costs only 0.0156 h to reduce the size of the ensemble state matrix) is little and ignorable. Note that the total computational time of $\mathbf{A}^a = \mathbf{A}^f \mathbf{X}$ with MS (i.e., 0.87 h in Table 2) is not exactly equal to the computational time of the MS-EnKF analysis procedures (i.e., 0.88 h in Table 1). The subtraction (i.e., 0.01 h) corresponds to the summed computational time of all the other analysis procedures (i.e., procedures 1 - 8) except for $\mathbf{A}^a = \mathbf{A}^f \mathbf{X}$ (see Fig. 2(b) and Table 3).

MS is now experimentally proven as efficient to significantly reduce the computational time for the analysis step during volcanic ash DA. Note that it can also be observed that the computational time for the "other" parts in Table 1 (such as operations for setting environmental variables, starting and finalizing DA algorithms, as mentioned in Sect. 3.1) is slightly reduced by the MS method (i.e., 0.03 h in this case). This is because in the conventional EnKF, the ensemble mean state $\bar{\mathbf{A}}^f$ is calculated in the "other" parts as an output to finalize the DA algorithms, while in MS-EnKF, the calculations of $\bar{\mathbf{A}}^f$ are needed and directly involved in the "Analysis" part.

The result shows that benefiting from the success of reduced analysis step, the overall computational cost indeed gets significantly reduced. The total execution time is 1.95 h which is less than the simulation window of 3 h (09:00 - 12:00 UTC, May 18, 2010). This result satisfies our goal to accelerate the computation to an acceptable runtime (i.e., requires less run time than the time period of the DA application). Therefore, aviation advices based on the MS-EnKF can be provided as not only accurate, but also sufficiently fast. Note that the result (1.95 h) is obtained after the volcanic ash is transported to the continental Europe. If the assimilation is performed in the starting phase of volcanic ash eruption (when aircraft measurements are available), a more significant acceleration would be obtained. This is because in this case the volcanic ash is only transported in an area near to the volcano, thus the number of no-ash grid cells will take a large proportion (much higher than $\frac{2}{3}$ for this case study) of the full domain.

There is another interesting point. According to Fig. 3, the ash grids comprises 39.3 % of the total grids. Thus, the minimum computing time by using MS to utilize this model's characteristic should be $\approx 1.234$ h (i.e., $0.393 \times 3.14$ h). However, the

experimental result shows that the computational time goes down to 0.88 h (see Table 1). One reason for this time decrease is that when the size of matrix is reduced, the memory access cost also goes down (e.g., through better cache usages). Another possible reason is that the ash grid number actually decreases with time (not always taking 39.3 % the total grid number), due to ash sedimentation and deposition processes ((Fu et al., 2015)).

Note that in this study, we only perform the commonly used ensemble parallelization for the forecast step (already efficient compared to the expensive analysis step), but do not choose model-based parallelization (e.g., tracer or domain decomposition). As specified in Table 1, no parallelization is implemented on the 6 tracers. This is because due to the important aggregation process (Folch et al., 2010), there are big dependencies between different ash components and thus it doesn't make much sense to parallelize them. As for domain-decomposed parallelization (Segers, 2002), it is not efficient for our application.

This is because volcanic ash is special in the sense that the model is only doing computations in a small part of the domain (i.e., there is no data in a rather large part of domain), and this active part is continuously changing. Thus, a fixed domain decomposition is not very useful here because of the changing plume position. In this sense, some advanced approach such as adaptive domain-decomposed parallelization (Lin et al., 1998) should be adopted to achieve additional acceleration to the volcanic ash forecast stage. This is an interesting subject for future application, when a more complicated model is employed,

only ensemble parallelization may be not enough for the forecast stage.

## 5   Comparison between MS and standard sparse matrix methods

### 5.1   Issues related to the generation of CSR-based arrays

According to Sect. 4, MS has proven to be capable of solving the computational issue of $\mathbf{A}^a = \mathbf{A}^f \mathbf{X}$. Motivated from model's characteristics, MS was proposed from an application's perspective and achieved a good result by managing the irregular

sparsity in our complicated volcanic ash DA system. The main reason why MS is efficient is that the sparsity of $\mathbf{A}^f$ can be well utilized by MS. In Sect. 4, we only performed the comparison between MS and the case of full storage dense matrices. However, the problem abstracted here ($\mathbf{A}^f \mathbf{X}$) is actually a sparse-dense matrix multiplication (SDMM) problem, since $\mathbf{A}^f$ is sparse and $\mathbf{X}$ is dense (see Eq. (12) for $\mathbf{X}$). Thus, one may wonder what the result would be, if the comparison of MS is made to more standard sparse matrix methods, such as Compressed Sparse Row (CSR)-based methods (Saad, 2003; Bank and

Douglas, 1993), which are commonly used for sparse matrix vector/matrix multiplication.

    Before we make the comparison, we need to first address the intrinsic problem when considering standard sparse matrix methods in EnKF for $\mathbf{A}^a = \mathbf{A}^f \mathbf{X}$. The issue is that it is not possible to directly generate a sparse storage format (e.g., CSR) of $\mathbf{A}^f$ without first generating the full matrix $\mathbf{A}^f$. This is mainly because $\mathbf{A}^f$ comes from the model-driven ensemble forecast step, where each $\mathbf{A}^f$ column corresponds to one member of the ensemble. During model forecast, we know there are indeed

no-ash grids. However, it is not sure where the plume exactly is after one forecasting time step. This is highly dependent on the weather conditions and the model processes (e.g, advection and diffusion for horizontal grids, sedimentation and deposition for vertical grids). Thus, a fixed and wide domain is usually needed by the model to avoid complications, resulting in the generation of the full storage of $\mathbf{A}^f$ (to be used in $\mathbf{A}^a = \mathbf{A}^f \mathbf{X}$). Therefore, if we want to implement a CSR storage format for

the sparse matrix $\mathbf{A}^f$, we must first generate the full storage $\mathbf{A}^f$ from the ensemble forecast step, and then we generate the three CSR arrays based on $\mathbf{A}^f$.

Generating CSR arrays is usually much more expensive (computationally) than a single sparse matrix-vector multiplication (SpMV). Thus, if we generate CSR arrays for only performing one-time SpMV, it would be meaningless from HPC's point of view. Fortunately, this is not the case for $\mathbf{A}^a = \mathbf{A}^f\mathbf{X}$ (i.e., SDMM), which can actually be considered as N-times SpMV. (Here, $\mathbf{X}$ has N columns, and one SpMV means the multiplication of $\mathbf{A}^f$ with one column of $\mathbf{X}$.) Thus, CSR-based SDMM might also be a candidate in reducing the computation time of $\mathbf{A}^a = \mathbf{A}^f\mathbf{X}$. It remains interesting to compare the performance of CSR-based SDMM and MS, in dealing with $\mathbf{A}^a = \mathbf{A}^f\mathbf{X}$ for our study case.

## 5.2   Result of CSR-based SDMM

To implement CSR-based SDMM for $\mathbf{A}^a = \mathbf{A}^f\mathbf{X}$, the three CSR arrays for $\mathbf{A}^f$ (denoted $\boldsymbol{val}$, $\boldsymbol{col\_idx}$, $\boldsymbol{row\_ptr}$ in this study) need to be first generated. The array $\boldsymbol{val}$ of size $n_{\boldsymbol{val}}$ stores non-zero values of $\mathbf{A}^f$, where $n_{\boldsymbol{val}} \approx n_{ash} N$. (In this study case, $n = 3.888 \times 10^6$, $n_{ash} \approx \frac{1}{3}n$, $N = 100$.) The array $\boldsymbol{col\_idx}$ of the same size $n_{\boldsymbol{val}}$ stores column index of the non-zeros. The array $\boldsymbol{row\_ptr}$ saves the start and the end pointers of the non-zeros of the rows in $\mathbf{A}^f$. The size of $\boldsymbol{row\_ptr}$ is $n+1$.

After the above three CSR arrays are generated, CSR-based SpMV can be performed for multiplying $\mathbf{A}^f$ with a column vector $\boldsymbol{v}$ in $\mathbf{X}$ (see Algorithm 1 in Fig. 4(a)). After that, Algorithm 2 (Fig. 4(b)) can be implemented by looping Algorithm 1 for $N$ times to obtain $\mathbf{A}^a = \mathbf{A}^f\mathbf{X}$. The experimental result of CSR-based SDMM is shown in Table 4, where all the environmental conditions (such as the DA system, the programming environment) are the same as the case of MS. This gives a fair comparison between CSR-based SDMM and MS. In addition, for a pure algorithmic comparison with the serial MS, here the CSR-based SDMM is also performed in a serial case.

From Table 4, we can first confirm that the computational time (i.e., 0.0407 h) for the generation of the three CSR-based arrays ($\boldsymbol{val}$, $\boldsymbol{col\_idx}$, $\boldsymbol{row\_ptr}$ to represent the sparse matrix $\mathbf{A}^f$) indeed takes more time than the computational time of one CSR-based SpMV (i.e., 0.0117 h). Thus, there is little value to perform sub-step (i) (see Table 4) if only one SpMV (i.e., sub-step (ii)) is needed. However, to get all $N$ (i.e., 100) columns of $\mathbf{A}^a$, the sub-step (ii) is looped for $N$ times, resulting in an ignorable impact of sub-step (i) on the total computational time (i.e., 1.21 h) of CSR-based SDMM.

The result of CSR-based SDMM also shows that the standard sparse matrix methods can reduce the computational time of $\mathbf{A}^a = \mathbf{A}^f\mathbf{X}$, by comparing with the conventional way in Table 3. However, it can also be observed that the computational time of CSR-based SDMM is larger than MS (i.e., 1.21 h versus 0.87 h in Table 3). Thus, although application of sparse matrix multiplication methods is positive, it is still slower than MS on our problem.

## 5.3   Comparison between CSR-based SDMM and MS

In the CSR-based SDMM, only non-zero elements in $\mathbf{A}^f$ participate in the multiplication between $\mathbf{A}^f$ and $\mathbf{X}$, thus redundant computation (related to zero elements in $\mathbf{A}^f$) is avoided. So the computation time of $\mathbf{A}^f\mathbf{X}$ is reduced with CSR-based SDMM. In the following, we analyze the performance difference between CSR-based SDMM and MS.

Firstly, from the programming's perspective, in CSR-based SDMM, the loop number for the rows of $\mathbf{A}^f$ is from 1 to $n$ (see Fig. 4(a)), while the corresponding loop number in MS is from 1 to $n_{ash}$ (see the step (iv) of MS in Sect. 4.2, $n_{ash} \approx \frac{1}{3}n$). Although only non-zero elements are used in the multiplication in CSR-based SDMM, the length of the outer loop is still $n$ (much larger than $n_{ash}$) which is the essential reason that MS is faster than CSR-based SDMM. Note that as discussed in Sect. 4.1, there are many zero rows in $\mathbf{A}^f$, thus CSR-based SDMM actually does nothing when it comes to a zero row, but still needs to execute the loop. Within each loop number, it has to check the information from $\boldsymbol{row\_ptr}$ (size $n+1$), where the value corresponding to a zero row is usually set to be the value in $\boldsymbol{row\_ptr}$ corresponding to the first following non-zero row.

Secondly, with respect to the algorithm, CSR-based SDMM utilizes the sparsity of $\mathbf{A}^f$ by its generation of three CSR arrays. While, MS not only utilizes the sparsity information of the sparse matrix $\mathbf{A}^f$, but also utilizes the consistency of ensemble forecasts, that is, ensemble forecasted states are not consistent in values but usually consistent in non-zero locations. This is a typical property in ensemble-based DA, resulting in that $N$ ensemble plumes are different in concentration values, but have similar transport directions/shapes (see Sect. 4.2). Thus, most of the zero elements in $\mathbf{A}^f$ are actually in zero rows of $\mathbf{A}^f$ for an EnKF application, which leads to a small number of non-zero rows ($n_{ash}$) compared to the full number of rows ($n$) of $\mathbf{A}^f$. Therefore, the way only considering $n_{ash}$ rows in $\tilde{\mathbf{A}}^f_{n_{ash} \times N} \mathbf{X}_{N \times N}$ (see the step (iv) of MS in Sect. 4.2) is more advantageous for an EnKF application than that considering all $n$ rows in CSR-based SDMM. Based on the above analysis, MS can be considered as a specific sparse matrix method, which typically works for ensemble-based DA applications.

It is useful to apply standard sparse matrix methods (e.g., CSR-based SDMM) for our assimilation application. The accelerated analysis step by CSR-based SDMM (1.22 h, see Table 3) also reduces the total computational time (i.e., 2.29 h, see Table 1 for the computational time of initialization, forecast and others) to an acceptable level (i.e., less than 3 h for our case study). In practice, due to the better performance of MS than CSR-based SDMM, we will use MS as a better choice for assimilation applications. In addition, we do not only intend to present MS, but also intend to reveal which part is the most time-consuming part for plume-type assimilation of in situ observations.

## 6   Discussions on MS

### 6.1   Applicability

For volcanic ash forecasts, only a relatively small domain is polluted compared to the full 3-D domain, so that MS can work efficiently. Using MS is also applicable for many other DA problems, where the domain is not fully polluted by the species. It does not matter what the emission looks like and whether the releases are short- or long-lived species. Given an assimilation problem, the only restriction for MS to gain an acceleration is whether the whole domain is fully polluted or partly polluted. The assimilation problems where MS can achieve the acceleration effect on the computations of $\mathbf{A}^a = \mathbf{A}^f \mathbf{X}$ include all the volcanic-related ash/gas assimilation, e.g., assimilation of satellite data/LIDAR data/in situ data; (sand/desert) dust storm related assimilation; tornado-related assimilation; assimilation of exploding nuclear plants or factories; chemicals or oils leaking on seas; global (forecast) fire assimilation; assimilation of environmental pollutant transport, e.g., severe smog. In addition, for DA applications (e.g., Ozone, $SO_2$) where pollutants spread over the whole domain, usually the focus is only on the high

concentrations and a threshold can be set to ignore the very low values without losing the necessary assimilation accuracy. In this case, MS can also lead to a potential acceleration since many very low concentrations can be explicitly truncated to be zeros.

It has been analyzed that when the number of non-zero rows ($n_{ash}$, i.e., the number of ash grids in a 3-D domain) of $\mathbf{A}^f$ satisfies $n_{ash} < n$, MS can work faster than standard EnKF. For volcanic ash application, because $n_{ash}$ is much less than $n$, the acceleration is thus quite large. Hence in this case, we propose to embed MS in all ensemble-based DA methods because it is fast and the implementation using MS is exact to the standard ensemble-based methods, i.e., it doesn't introduce any approximation in view of MS procedures. Actually this proposal can be extended to all real applications, even if the condition is not satisfied. This is because, in this case the computational cost of MS for $\mathbf{A}^a = \mathbf{A}^f \mathbf{X}$ becomes $O(n N^2)$, which is the same as that of using the standard assimilation (shown in Fig. 2(b)). Therefore, if the state numbers equal to or close to the number of the total number of grid points in the domain, the added computational cost by using MS is very small (neglectable), so that the computational time with MS is almost the same as the time of using the standard approach. Whereas, when the condition $n_{ash} < n$ is satisfied, MS will accelerate the analysis step. Thus MS is generic and can be directly used in any ensemble-based DA, and this acceleration can be automatically realized for some potential applications, without spending time investigating if the condition is satisfied. In a real (or operational) 3-D DA system, MS can be easily included, i.e., we only need to invoke the MS module when computing $\mathbf{A}^a = \mathbf{A}^f \mathbf{X}$, without any other change to the current framework.

As stated in Eq. (15), the speedup of the MS method is approximately the inverse of $\frac{n_{ash}}{n}$. So far there is no statistical data on the value of $\frac{n_{ash}}{n}$. Considering the problem of volcanic ash transport, there is one emission point (at the volcano), all the ashes in atmospheres are transported by the directional wind drive from the same source point. Thus volcanic ash cloud is actually transported in a shape of a plume, which in general doesn't cover the full but only a small part of the 3-D domain. At the start phase of a volcanic ash eruption, $\frac{n_{ash}}{n}$ is much smaller than 1.0 (started from 0). During transport over a long time (one and a half months for this case study), $\frac{n_{ash}}{n}$ increases to approximately $\frac{1}{3}$. Therefore, the speedup of MS on volcanic ash DA will be significant.

### 6.2 MS and localization

Based on the formulation of MS, one may think it can be taken as a localization approach ((Hamill et al., 2001)). There is indeed a similarity between MS and the localization approach, in a sense that when computing $\mathbf{A}^a = \mathbf{A}^f \mathbf{X}$, both get rid of a large number of cells, and only do computations related to the selected grids. These two algorithms are however functionally different. This is because the localization approach is meant for reducing spurious covariances outside a local region which is built up around the measurement, thus the results with and without localization approaches are different. While, MS is developed for the acceleration purpose. The masked region is discontinuous and independent of locations of measurement, but dependent on the model domain. Thus, there is no difference on the assimilation results between using MS and without using it. Therefore, based on the functional difference, MS cannot be taken as a localization approach.

In this study, we don't employ the localization strategy in the analysis step, because we use a rather large ensemble size of 100 to guarantee the accuracy, as introduced in Sect. 2. But for some applications (e.g., ozone, $CO_2$, sulfur dioxide) especially

when assimilating satellite data, localization is a necessary approach and has been widely used in reducing spurious covariances (Barbu et al., 2009; Chatterjee et al., 2012; Curier et al., 2012). In these cases, because the localization approach forces the analysis only to update state within a localization region, one may think that localization could replace MS and there would be no significance to employ MS. Actually this is not correct. We explain the reason as follows.

The localization approach is usually realized in Eq. (6) by employing a Schur product of a localization matrix and the forecast error covariance matrix (Houtekamer and Mitchell, 1998, 2001) given by:

$$\mathbf{K}(k) = (\mathbf{f} \circ \mathbf{P}^f(k))\mathbf{H}(k)^T[\mathbf{H}(k)(\mathbf{f} \circ \mathbf{P}^f(k))\mathbf{H}(k)^T + \mathbf{R}]^{-1}. \tag{17}$$

The Schur product $\mathbf{f} \circ \mathbf{P}^f$ in Eq. (17) is defined by the element-wise multiplication of the covariance matrix $\mathbf{P}^f$ and a localization matrix $\mathbf{f}$. $\mathbf{f}$ is defined based on the distance between two locations, thus it is dependent on the domain and needs
information of the full ensemble state locations. In this way, $\mathbf{f} \circ \mathbf{P}^f$ can contain more zeros than $\mathbf{P}^f$, but the dimensions are not changed, so that the computations related to $\mathbf{f} \circ \mathbf{P}^f$ are actually not reduced. Therefore, we can understand the localization approach in the analysis step as that the state within and outside a local region are both updated with increments, but just the increments outside the region are zero (which seems like not updating). This is also the reason why the localization approach is not meant for acceleration but only for reducing spurious covariances. Now it is clear that localization cannot replace MS.
Actually both can be performed together in dealing with the time-consuming part $\mathbf{A}^a = \mathbf{A}^f \mathbf{X}$. The localization approach can first transfer $\mathbf{A}^f$ to a localized matrix with more zero rows. Then MS can be used to accelerate the multiplication of the localized matrix and $\mathbf{X}$. In this way, MS is expected to accelerate $\mathbf{A}^a = \mathbf{A}^f \mathbf{X}$ with a high speedup rate, because the computational cost of more zero rows in the localized ensemble state matrix are avoided.

### 6.3   MS and parallelization

Motivated from the model's physics, the implementation MS currently is for the serial case. This implementation has reduced the computation time to an acceptable time (i.e., simulation time is less than the period of forecast in real world time). It is however interesting to discuss the potential of parallelization of the dense-dense matrix multiplication ($\tilde{\mathbf{A}}^a_{n_{ash} \times N} = \tilde{\mathbf{A}}^f_{n_{ash} \times N} \mathbf{X}_{N \times N}$) in the step (iv) of the algorithm (see Sect. 4.2 and Table 2). The related matrix multiplication can be easily parallelized on multiple processors. Optimization and evaluation on the parallelized MS will be considered in future. For the
current case study, the computational time (3.13 h, see Table 3) for an "ideal" reduction by parallelization of MS is not much larger than the acceleration (already) gained by MS (2.26 h, subtraction between 3.13 h and 0.87 h, see Table 3). Therefore, from application's perspective, further acceleration by parallelization is not required.

Alternatively, one may also consider to (1) directly parallelize the expensive matrix multiplication of $\mathbf{A}^a_{n \times N} = \mathbf{A}^f_{n \times N}\mathbf{X}_{N \times N}$, without first performing MS; or (2) implement CSR-based SDMM (see Sect. 5) with parallelization. Both are possible alterna-
tive approaches to accelerate the expensive matrix multiplication. The first approach can be implemented by user own designed parallelization, or by utilizing scaLAPACK (https://www.netlib.org/scalapack/, where the main function is "pdgemm"). The second approach can be realized by using some general parallel sparse-dense matrix multiplication methods (e.g., sending each column of X and three CSR arrays of $\mathbf{A}^f$ to each processor to calculate each column of $\mathbf{A}^a$) or using a good parallel alge-

bra library like PeTSC (https://www.mcs.anl.gov/petsc/) which allows users to specify own orderings and comes with machine optimized parallel matrix-matrix multiplication operations. However, given the fact that MS can also be parallelized using similar ways or same libraries, thus it is fair to not consider parallelization for all cases (i.e., using MS, not using MS, using CSR-based SDMM). Actually, the parallelization in MS could be performed much easier than other approaches in dealing with

$\mathbf{A}^a = \mathbf{A}^f \mathbf{X}$, because the dense-dense matrix multiplication (parallelization in the step (iv) of MS) is easier to parallelize than the sparse-dense matrix multiplication (direct parallelization for $\mathbf{A}^a = \mathbf{A}^f \mathbf{X}$ or parallelized CSR-based SDMM).

In this paper, for the current usage, we keep the possibility of parallelization open, because a serial MS has been efficient already.

## 7 Conclusions

In this study, based on evaluations on the computational cost of volcanic ash DA, the analysis step turned out to be very expensive. Although some potential approaches can accelerate the initialization and forecast step, there would be no notable improvement to the total computational time due to the dominant analysis step. Therefore, to get an acceptable computational cost, the key is to efficiently reduce the execution time of the analysis step.

After a detailed evaluation on various parts of the analysis stage, the most time-consuming part was revealed. The mask-state

algorithm (MS) was developed based on a study on the characteristic of the ensemble ash states. The algorithm transforms the full ensemble state matrix into a relatively small matrix using a constructed mask array. Subsequently, the computation of the analysis step was sufficiently reduced. MS is developed as a generic approach, thus it can be embedded in all ensemble-based DA implementations. The extra computational cost of the algorithm is small and usually neglectable.

The conventional ensemble-based DA with MS is shown to successfully reduce the total computational time to an acceptable

level, i.e., less than the time period of the assimilation application. Consequently, timely and accurate volcanic ash forecasts can be provided for aviation advices. This approach is flexible. It boosts the performance without considering any model-based parallelization, such as domain or component decomposition. Thus, when a parallel model is available, MS can be easily combined with the model to gain a further speedup. It implements exactly the standard DA without any approximation and with easy configurations, so that it can be used to accelerate the standard DA in a wide range of applications.

In this case study with the LOTOS-EUROS model (version 1.10), after the parallelization is performed for the forecast step of EnKF assimilation, the analysis step takes 72 % of total runtime, which means the analysis step is the bottle neck. This case might not be general for all ash forecasts. As the computational cost for initialization and forecast greatly depends on the forecast model that is used. For the current development, it makes sense to use the LOTOS-EUROS model, because the model has been configured and evaluated in (Fu et al., 2015) by comparison with other famous models (e.g., NAME (Jones et al.,

2007), WRF-Chem (Webley et al., 2012)) in simulating volcanic ash transport. However, if a more expensive ash forecasting model is used, then the bottle neck would be the forecast step. In this case, the forecast step should be the goal for acceleration and probably a parallel model or adaptive domain-decomposition (as discussed in Sect. 4.3) needs to be employed together with the parallel ensemble forecasts.

The use of in situ measurements is one important reason why MS perfectly works. For each analysis step, the number of measurements are quite small, and the procedure of the singular value decomposition (SVD) costs little. However, in some applications when many measurements are assimilated (e.g., satellite-based or seismic-based data), and the number of measurements is of the same order as the number of state variables, the most time-consuming part will be the SVD. In these cases, the contributions of MS will be limited. The reduction of the total computing time using MS therefore is less significant, an effective acceleration algorithm for the analysis step must be used and should consider the computationally-expensive SVD in the first place.

## 8 Data and code availability

The averaged aircraft in situ data used in this study are available from Fig. 1(c). The used continuous aircraft data and the model output data can be accessed by request (G.Fu@tudelft.nl). The mask-state algorithm (MS) is implemented in OpenDA (the open source software for DA, www.openda.com) and the software can be downloaded from sourceforge (https://sourceforge.net/projects/openda).

*Author contributions.* Guangliang Fu, Sha Lu and Arjo Segers simulated the volcanic ash transport using the LOTOS-EUROS model. Guangliang Fu, Hai Xiang Lin, Tongchao Lu evaluated the computational efforts. Guangliang Fu, Hai Xiang Lin, Arnold Heemink and Shiming Xu developed the algorithms. Guangliang Fu, Hai Xiang Lin and Nils van Velzen carried out computer experiments and analyzed the performance of the developed algorithm. Guangliang Fu and Hai Xiang Lin wrote the paper.

5   *Competing interests.* The authors declare that they have no conflict of interest.

*Acknowledgements.* We are very grateful to the editor and reviewers for their reviews and insightful comments. We thank the Netherlands Supercomputing Center to support us the "Cartesius" cluster for the experiments in our study. We are grateful to Konradin Weber for providing the aircraft measurements.

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

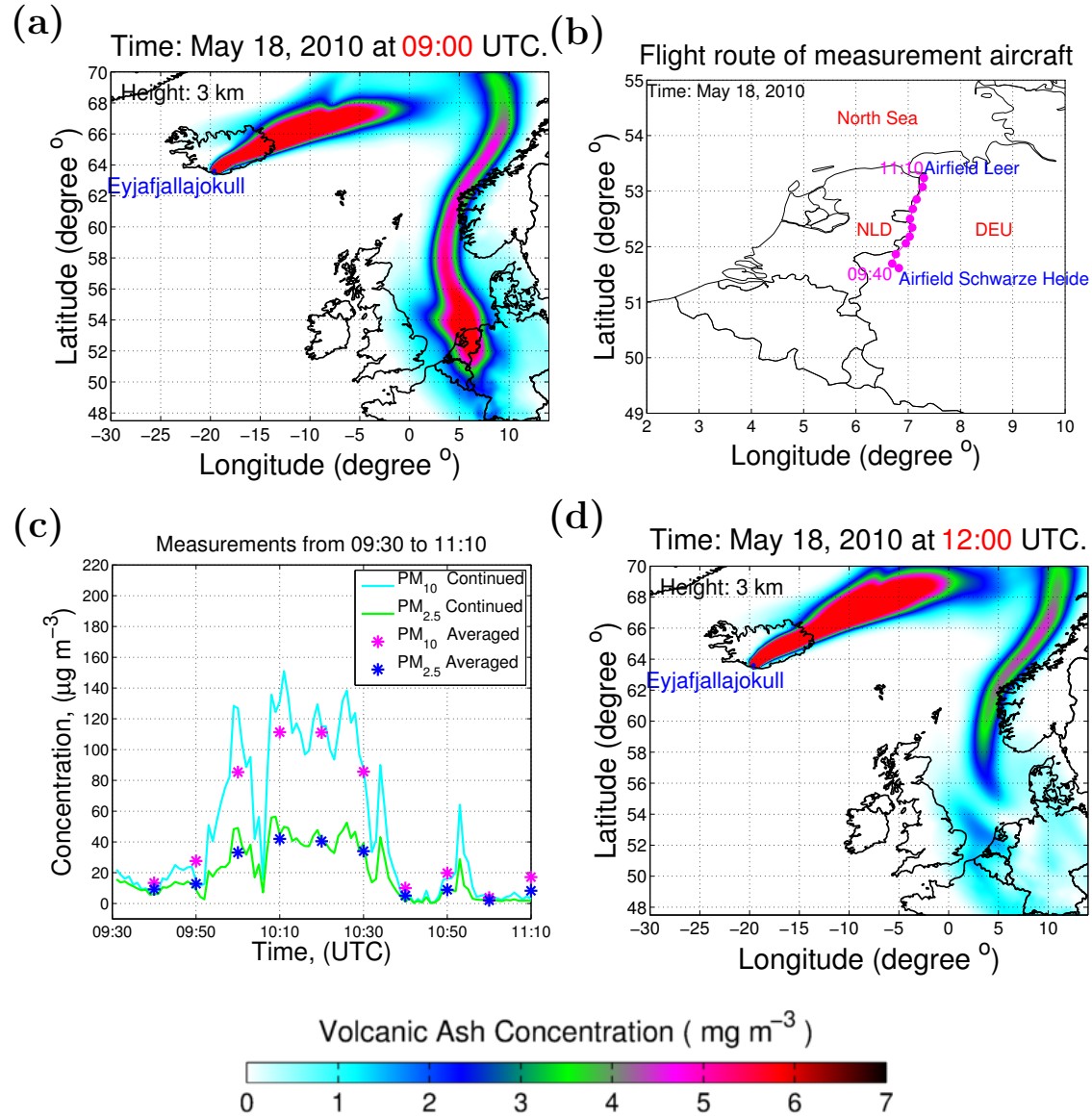

**Figure 1. Methodology of ensemble-based DA. (a)** The initial volcanic ash state at 09:00 UTC. **(b)** Flight route of measurement aircraft. **(c)** Aircraft in situ measurements of $PM_{10}$ and $PM_{2.5}$ from 09:30 to 11:10 UTC May 18, 2010. **(d)** Volcanic ash assimilation result at 12:00 UTC.

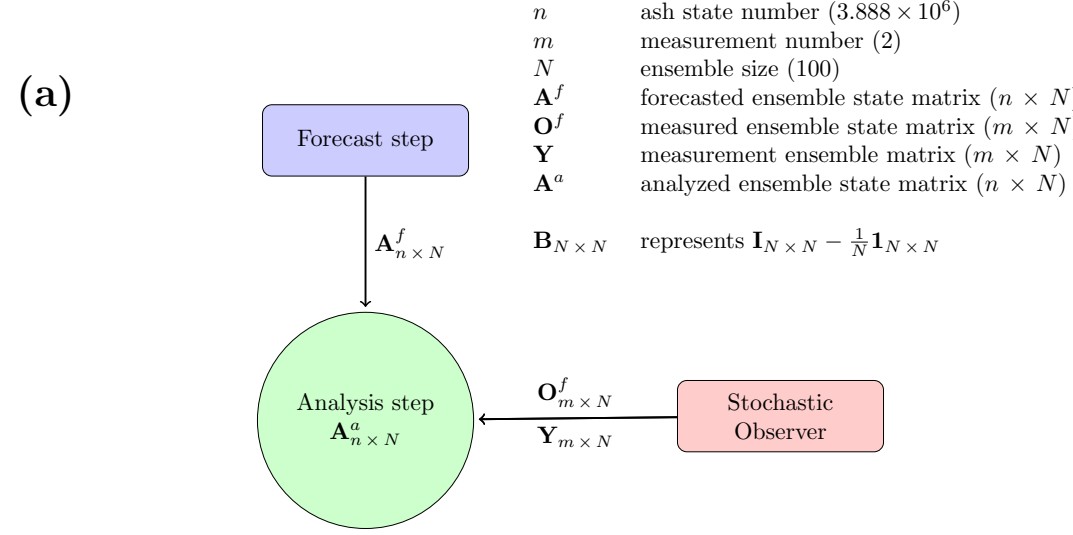

**(a)**

| | |
|---|---|
| $n$ | ash state number $(3.888 \times 10^6)$ |
| $m$ | measurement number $(2)$ |
| $N$ | ensemble size $(100)$ |
| $\mathbf{A}^f$ | forecasted ensemble state matrix $(n \times N)$ |
| $\mathbf{O}^f$ | measured ensemble state matrix $(m \times N)$ |
| $\mathbf{Y}$ | measurement ensemble matrix $(m \times N)$ |
| $\mathbf{A}^a$ | analyzed ensemble state matrix $(n \times N)$ |
| $\mathbf{B}_{N \times N}$ | represents $\mathbf{I}_{N \times N} - \frac{1}{N}\mathbf{1}_{N \times N}$ |

**(b)**

**Computational cost of analysis step**

| Procedures | Cost |
|---|---|
| 1. $\mathbf{X_1} = \mathbf{O}^f\mathbf{B}$ | $\mathrm{O}(mN^2)$ |
| 2. $\mathbf{X_2} = \mathbf{YB}$ | $\mathrm{O}(mN^2)$ |
| 3. $\mathbf{X_3} = \mathbf{X_1}\mathbf{X_1}' + \mathbf{X_2}\mathbf{X_2}'$ | $\mathrm{O}(m^2N)$ |
| 4. $\mathbf{X_4} = \mathbf{X_3}^{-1}$ (Singular Value Decomposition (SVD)) | $\mathrm{O}(m^3)$ |
| 5. $\mathbf{X_5} = \mathbf{B}\mathbf{X_1}'$ | $\mathrm{O}(mN^2)$ |
| 6. $\mathbf{X_6} = \mathbf{X_5}\mathbf{X_4}$ | $\mathrm{O}(m^2N)$ |
| 7. $\mathbf{X} = \mathbf{I} + \mathbf{X_6}(\mathbf{Y} - \mathbf{O}^f)$ | $\mathrm{O}(mN^2)$ |
| 8. $\mathbf{A}^a = \mathbf{A}^f\mathbf{X}$ | $\mathrm{O}(nN^2)$ |

$(n=3.888 \times 10^6,\ m=2,\ N=100.)$

**Figure 2. Computational evaluation of the analysis step.** **(a)** Illustration of the analysis step. **(b)** Computational cost of all sub-part of the analysis step.

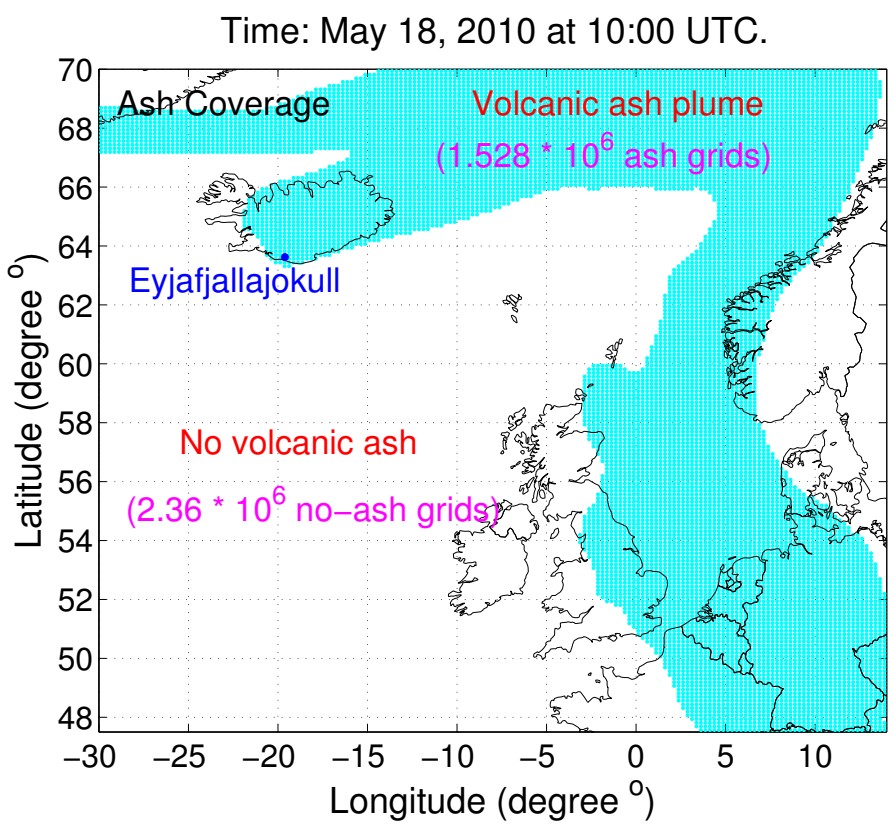

**Figure 3. Characteristic of volcanic ash state.**

**(a)**

CSR-based SpMV ($\mathbf{A}^f$ is represented in CSR format)

| **Algorithm 1** $\quad \mathbf{A}^a[:,1]_{n \times 1}$=CSR-based SpMV($\mathbf{A}^f_{n \times N}$, $\boldsymbol{v}_{N \times 1}$) |
|---|
| 1.    for $i = 1 : n$ |
| 2.        $\mathbf{A}^a[i,1] = 0$ |
| 3.        for $j = \boldsymbol{row\_ptr}[i] : \boldsymbol{row\_ptr}[i] - 1$ |
| 4.           $w = \boldsymbol{col\_idx}[j]$ |
| 5.           $\mathbf{A}^a[i,1] \mathrel{+}= \boldsymbol{val}[j] \times \boldsymbol{v}[w]$ |
| 6.        end for |
| 7.    end for |

$(n=3.888 \times 10^6,\ N=100.)$

**(b)**

SDMM is conducted by looping SpMV $N$ times

| **Algorithm 2** $\quad \mathbf{A}^a[:,1:N]_{n \times N} = N$-times **Algorithm 1** ) |
|---|
| 1.    for $k = 1 : N$ |
| 2.        $\mathbf{A}^a[:,k]_{n \times 1} = $ CSR-based SpMV($\mathbf{A}^f_{n \times N}$, $\mathbf{X}[:,k]_{N \times 1}$) |
| 3.    end for |

$(n=3.888 \times 10^6,\ N=100.)$

**Figure 4. Algorithms for CSR-based SDMM to compute the multiplication of sparse matrix $\mathbf{A}^f$ and dense matrix X. (a)** Multiplication of $\mathbf{A}^f$ with a column vector $\boldsymbol{v}$ (in $\mathbf{X}$) by CSR-based sparse matrix vector multiplication (SpMV). $\boldsymbol{val}$, $\boldsymbol{col\_idx}$, $\boldsymbol{row\_ptr}$ are the three arrays to represent $\mathbf{A}^f$ in CSR format. **(b)** Looping SpMV N times (each with one column of $\mathbf{X}$) to obtain $\mathbf{A}^a = \mathbf{A}^a\,\mathbf{X}$.

**Table 1.** Comparison of the computational cost of conventional EnKF and MS-EnKF. ( The results are obtained from the bullx B720 thin nodes of the Cartesius cluster, which is a computing facility of SURFsara, the Netherlands Supercomputing Centre. Each node is configured with $2 \times 12$-core 2.6 GHz Intel Xeon E5-2690 v3 (Haswell) CPUs and with memory 64 GB.)

| Case | Conventional EnKF | MS-EnKF |
|---|---|---|
| Cores used | 102 | 102 |
| Tracer number ($n_{spec}$) | 6 | 6 |
| Measurements of tracers (m) | 2 | 2 |
| Ensemble size (N) | 100 | 100 |
| Parallel in forecast step | Yes | Yes |
| Parallel in analysis step | No | No |
| Mask-state in analysis step | **No** | **Yes** |
| Initialization | 0.42 h | 0.42 h |
| Forecast | 0.65 h | 0.65 h |
| Analysis | **3.14** h | **0.88** h |
| Others | 0.15 h | 0.12 h |
| Total Runtime | **4.36** h | **1.95** h |

h=hour, simulation window = **3.0** h, the time is Wall clock time.

**Table 2.** Computational evaluation of all the steps of the mask-state algorithm (MS) for $\mathbf{A}^a = \mathbf{A}^f \mathbf{X}$. (see details of each step of MS in Sect. 4.2.)

| Sub-step | Computational time |
|---|---|
| (i) Compute ensemble mean state $\bar{\mathbf{A}}^f$ | 0.0097 h |
| (ii) Construct mask array $\mathbf{z}$ | 0.0002 h |
| (iii) Construct masked ensemble state matrix $\tilde{\mathbf{A}}^f$ | 0.0057 h |
| (iv) Compute $\tilde{\mathbf{A}}^a$ by multiplying $\tilde{\mathbf{A}}^f$ and $\mathbf{X}$ | **0.8474** h |
| (v) Construct analyzed ensemble state matrix $\mathbf{A}^a$ | 0.0070 h |
| Total | **0.87** h |

h=hour, the time is Wall clock time.

**Table 3.** Computational time for the analysis step of conventional EnKF, MS-EnKF, CSR-based-SDMM-EnKF.

| Analysis's procedures (see Fig. 2(b)) | Conventional EnKF | MS-EnKF | CSR-based-SDMM-EnKF |
|---|---|---|---|
| procedures 1 - 8 | 0.01 h | 0.01 h | 0.01 h |
| procedure 9 ($\mathbf{A}^a = \mathbf{A}^f \mathbf{X}$) | **3.13** h | **0.87** h | **1.21** h |
| Total | **3.14** h | **0.88** h | **1.22** h |

h=hour, the time is Wall clock time.

**Table 4.** Computational evaluation of the sub-steps of the Sparse-Dense Matrix Multiplication with Compressed Sparse Row storage (CSR-based SDMM) for $\mathbf{A}^a = \mathbf{A}^f \mathbf{X}$.

| Sub-step | Computational time |
|---|---|
| (i) Compute three arrays (in CSR format) of $\mathbf{A}^f$ | 0.0407 h |
| (ii) Compute CSR-based SpMV for the first column of $\mathbf{A}^a$ | 0.0117 h |
| (iii) Loop (ii) for N-1 times for other N-1 columns of $\mathbf{A}^a$ | **1.1576** h |
| Total | **1.21** h |

CSR-based SDMM is formed by (ii) and (iii). h=hour, the time is Wall clock time.