# Peer review of "Accelerating volcanic ash data assimilation using a mask-state algorithm based on ensemble Kalman filter: a case study with the LOTOS-EUROS model (version 1.10)"

_Geoscientific Model Development, 2016_

## Short Comment (SC1) · 13 Sep 2016

Dear authors,

In my role as Executive editor of GMD, I would like to bring to your attention our Editorial version 1.1:

http://www.geosci-model-dev.net/8/3487/2015/gmd-8-3487-2015.html

This highlights some requirements of papers published in GMD, which is also available on the GMD website in the 'Manuscript Types' section:

http://www.geoscientific-model-development.net/submission/manuscript_types.html

In particular, please note that for your paper, the following requirements have not been met in the Discussions paper:

[Figure]

- "The main paper must give the model name and version number (or other unique identifier) in the title."

- "If the model development relates to a single model then the model name and the version number must be included in the title of the paper. If the main intention of an article is to make a general (i.e. model independent) statement about the usefulness of a new development, but the usefulness is shown with the help of one specific model, the model name and version number must be stated in the title. The title could have a form such as, "Title outlining amazing generic advance: a case study with Model XXX (version Y)"."

In order to simplify reference to your developments, please add a model name (and/or its acronym) and a version number in the title of your article in your revised submission to GMD.

Yours,

Astrid Kerkweg

---

## Author Comment (AC1) · 13 Sep 2016

Dear Dr. Kerkweg (Executive Editor),

Thank you for pointing out the problem. We have changed the title to "A mask-state algorithm to accelerate volcanic ash data assimilation: a case study with the LOTOS-EUROS model (version 1.10)".

In addition, we have added the related information in Section 2: "To simulate a volcanic ash plume, an atmospheric transport model is needed. In this paper, the LOTOS-EUROS (abbreviation of LOng Term Ozone Simulation – EURopean Operational Smog) model is used (Schaap et al., 2008) with model version 1.10 (http://www.lotos-euros.nl/). The LOTOS-EUROS model (Schaap et al., 2008) is an operational model focusing on nitrogen oxides, ozone, particular matter, volcanic ash."

The changed manuscript is attached.

Best Regards, Guangliang Fu on behalf of all co-authors

Please also note the supplement to this comment:
http://www.geosci-model-dev-discuss.net/gmd-2016-208/gmd-2016-208-AC1-supplement.pdf

―――――――――――――――――――――

**Supplement:**

[revised manuscript text omitted]

---

## Referee Comment (RC1) · Anonymous Referee #1 · 24 Sep 2016

General comments:

The article presents a nice study of Ensemble Kalman Filter application for the prediction of volcanic ash and an extension for acceleration. Indeed, the analysis step involves large computational cost and this cost is reduced by employing the existence of "zero rows" in the ensemble matrix. As parallel computing is efficiently used (e.g. Khairullah et al. 2013) to reduce the analysis time in data assimilation and the authors also use parallel computing facilities only for the forecast step, which can be treated as an under utilization of the resources. Certainly inclusion of parallel computing (mentioned as a future work) also in the analysis step (both for the conventional and MS EnKF) would make the work more complete. Nevertheless, the reported work is an important research and is worth reading. Some improvements/clarifications/modifications are suggested below.

[Figure]

Specific comments:

1. The variable notation $N_{ash}$ should changed to $n_{ash}$ as the small n presents number of grid points (line 4.30).

2. Equation (15) indicates whether we would have speedup by the MS method (line 4.5). Could it be extended to explicitly specify the expected amount of speed up (with some assumptions if necessary)?

3. Some explanations are necessary about the involved operations in the others part for the clarification and justification of the less (by 0.03h) computational time (Table-1).

4. The computational overhead involved to transform the full matrix to a small one and vice versa should be explicitly discussed. The effect in the computation time should be also discussed.

5. Line 3.20 : It is mentioned that 72% of the total runtime is spent on the analysis step. This information can be further extended (preferably with mathematical formulation) to predict the maximum computational speedup (e.g. 3.57 for the case in this paper). c.f. Amdahl's law (Amdahl, Gene M. (1967). "Validity of the Single Processor Approach to Achieving Large-Scale Computing Capabilities". AFIPS Conference Proceedings (30): 483–485.)

6. Related to the above point, a further discussion about the discrepancy in the actual speedup (e.g. 2.24 for the case in this paper) from the predicted speedup could be included.

7. It is understood form equation (15) the total speedup is quite low, is there any statistical data regarding the value of $n_{ash}/n$ to know the probability of the applicability of the proposed method and justify the weight of the work?

8. The paper investigates actually a single method (MS) to accelerate volcanic ash data assimilation. Other methods are only referred (e.g. parallelization of the analysis step) or contrasted (e.g. localization) and are not really investigated in this paper. So,

the plural form (e.g. strategies) should be omitted.

---

## Author Comment (AC2) · 4 Oct 2016

Dear Anonymous Referee #1,

On behalf of all co-authors, first of all I would like to thank you for giving us very useful comments and suggestions. We really appreciate your detailed comments and suggestions, which certainly improve the quality of the paper. In the revision, we have carefully considered all the suggested changes and included them to the best of our ability.

In the following we will give our response to every comment. To make the changes easier to identify, we have numbered them.

Best regards,
Guangliang Fu
on behalf all co-authors

(The revised manuscript is in the latter part of this pdf.)

**Reply to Specific comments**:

1. *The variable notation $N_{ash}$ should be changed to $n_{ash}$ as the small n presents number of grid points (line 4.30).*

   Response: I agree. All $N_{ash}$ have been changed to $n_{ash}$ in the revision.

2. *Equation (15) indicates whether we would have speedup by the MS method (line 4.5). Could it be extended to explicitly specify the expected amount of speed up (with some assumptions if necessary)?*

   Response: As suggested, it is extended in line(s) P9L29–P9L33 (P:page, L:lines):
   " The relation $n_{ash} < \frac{N-1}{N}n$ indicates whether we would have speedup by the MS method, actually it can be extended to Eq. (15),

$$S_{\mathbf{ms}} = \frac{\mathbf{O}(nN^2)}{\mathbf{O}(nN + n_{ash}N^2)} = \mathbf{O}(\frac{n}{n_{ash}}), \tag{15}$$

   which explicitly specifies the expected amount of speedup ($S_{\mathbf{ms}}$) of $\mathbf{A^a} = \mathbf{A^f X}$ by the MS algorithm. In this case study, $N$ is taken at 100 and $\frac{n_{ash}}{n} \approx \frac{1}{3}$, so $S_{\mathbf{ms}}$ is approximately 3.0. "

3. *Some explanations are necessary about the involved operations in the others part for the clarification and justification of the less (by 0.03h) computational time (Table-1).*

   Response: I agree. The explanations are added in line(s) P10L22–P10L26:
   " Note that it can also be observed that the computational time for the "other" parts in Table 1 (such as operations for setting environmental variables, starting and finalizing data assimilation algorithms, as mentioned in Section 3.1) is slightly reduced by the MS method (i.e., 0.03 h in this case). This is because in the conventional EnKF, the ensemble mean state $\mathbf{\bar{A}^f}$ is calculated in the "other" parts as an output to finalize the data assimilation

algorithms, while in MS-EnKF, the calculations of $\bar{\mathbf{A}}^{\mathbf{f}}$ are needed and directly involved in the "Analysis" part. "

4. *The computational overhead involved to transform the full matrix to a small one and vice versa should be explicitly discussed. The effect in the computation time should be also discussed.*

   Response: As suggested, the computational overhead involved to transform the full matrix to a small one and vice versa are explicitly discussed in line(s) P8L17–P8L19:
   " **Construct masked ensemble state matrix** $\tilde{\mathbf{A}}^{\mathbf{f}}$: Using the mask array $\mathbf{z}_{n_{ash} \times 1}$ obtained from step (ii), $\tilde{\mathbf{A}}^{\mathbf{f}}_{n_{ash} \times N}$ can be constructed column by column according to Eq. (13), and the computational cost (overhead) for this step is $O(n_{ash}N)$.

   $$\tilde{\mathbf{A}}^{\mathbf{f}}(1 : n_{ash}, 1 : N) = \mathbf{A}^{\mathbf{f}}(\mathbf{z}(1 : n_{ash}), 1 : N), \tag{13}$$

   "

   and in line(s) P8L24–P8L27:
   " **Construct analyzed ensemble state matrix** $\mathbf{A}^{\mathbf{a}}$: With the computed $\tilde{\mathbf{A}}^{\mathbf{a}}$ from step (iv) and the mask array $\mathbf{z}$ from step (ii), the final analyzed ensemble state matrix $\mathbf{A}^{\mathbf{a}}_{n \times N}$ can be constructed based on Eq. (14). The computational cost (overhead) for this step is $O(nN)$.

   $$\mathbf{A}^{\mathbf{a}}(\mathbf{z}(1 : n_{ash}), 1 : N) = \tilde{\mathbf{A}}^{\mathbf{a}}(1 : n_{ash}, 1 : N), \tag{14}$$

   "

   The effect in the computation time is discussed in line(s) P9L15–P9L20:
   " Now we analyze MS's computational cost, which can be summed as $O(nN)+ O(n)+ O(n_{ash}N)+ O(n_{ash}N^2)+ O(nN)$, i.e., $O(nN + n_{ash}N^2)$. Thus, the computational overhead involved to transform the full matrix to a small one (i.e., $O(n_{ash}N)$ for procedure (iii)) has little effect in the total computation cost of MS (i.e., $O(nN + n_{ash}N^2)$). However, the computational overhead of transforming the small matrix to the full one (i.e., $O(nN)$ for procedure (v)) does contribute a part, which cannot be ignored, to the total MS's computational cost. "

5. *Line 3.20 : It is mentioned that 72% of the total runtime is spent on the analysis step. This information can be further extended (preferably with mathematical formulation) to predict the maximum computational speedup (e.g. 3.57 for the case in this paper). c.f. Amdahl's law (Amdahl, Gene M. (1967). "Validity of the Single Processor Approach to Achieving Large-Scale Computing Capabilities". AFIPS Conference Proceedings (30): 483-485.)*

6. *Related to the above point, a further discussion about the discrepancy in the actual speedup (e.g. 2.24 for the case in this paper) from the predicted speedup could be included.*

   Response: I agree. An extended discussion is added in line(s) P10L1–P10L9:
   " According to Amdahl's law (Amdahl, 1967), the total computational speedup ($S_{\mathbf{total}}$)

by MS can be predicted by Eq. (16),

$$S_{\textbf{total}} = \frac{1}{(1 - p_{\textbf{ms}}) + \frac{p_{\textbf{ms}}}{S_{\textbf{ms}}}}, \tag{16}$$

where $p_{\textbf{ms}}$ is the proportion of the computational cost of $\mathbf{A^a} = \mathbf{A^f X}$ in the overall data assimilation computations. It has been evaluated that the computational cost of $\mathbf{A^a} = \mathbf{A^f X}$ dominates the analysis step (see Fig. 2**b**), thus the proportion of the computational cost of $\mathbf{A^a} = \mathbf{A^f X}$ approximates the proportion of the analysis step in the total data assimilation computations (i.e., $p_{\textbf{ms}} \approx 72\%$ in this case, as described in Section 3.1). Therefore, based on Eq. (16), the maximum ("ideal") computational speedup can be predicted to be $\frac{1}{1-p_{\textbf{ms}}}$ (i.e., $\approx 3.57$ for this case study) when $S_{\textbf{ms}}$ approximates infinity. However, this is not the actual speedup because $S_{\textbf{ms}}$ is in fact specified by Eq. (15). (Based on discussions above, $S_{\textbf{total}}$ can be therefore estimated by Eq. (15) at $\approx 2.0$ in this case.) "

7. *It is understood from equation (15) the total speedup is quite low, is there any statistical data regarding the value of $n_{ash}/n$ to know the probability of the applicability of the proposed method and justify the weight of the work?*

Response: Thanks for the suggestion. The related discussion is added in line(s) P12L1–P12L7:
" As stated in Eq. (15), the speedup of the MS method is approximately the inverse of $\frac{n_{ash}}{n}$. So far there is no statistical data on the value of $\frac{n_{ash}}{n}$. Consider the problem of volcanic ash transport, there is only one emission point (at the volcano), all the ashes in atmospheres are transported by the directional wind drive from the same source point. Thus volcanic ash cloud is actually transported in a shape of a plume, which in general doesn't cover the full but only a small part of the 3D domain. At the start phase of a volcanic ash eruption, $\frac{n_{ash}}{n}$ is much smaller than 1.0 (started from 0). During transport over a long time (one and a half months for this case study), $\frac{n_{ash}}{n}$ increases to approximately $\frac{1}{3}$. Therefore, the speedup of MS on volcanic ash data assimilation will be significant. "

8. *The paper investigates actually a single method (MS) to accelerate volcanic ash data assimilation. Other methods are only referred (e.g. parallelization of the analysis step) or contrasted (e.g. localization) and are not really investigated in this paper. So, the plural form (e.g. strategies) should be omitted.*

Response: I agree. All "strategies" in the plural form have been changed to "a strategy" in the revision.

The revised manuscript starts from next page.

[revised manuscript text omitted]

---

## Referee Comment (RC2) · Anonymous Referee #2 · 22 Nov 2016

Referee report for the paper by G. Fu et al, called "A mask-state algorithm to accelerate volcanic ash data assimilation".

After reading the manuscript in detail I found the scientific content of the paper very minimal, and as such I am not in favour of publication in GMD. The authors published the ash assimilation approach and results in an earlier paper (Fu et al., ACP 2016). Compared to this earlier work I obtained very little extra knowledge from this new paper.

The main purpose of the paper is to show how the computational speed of the volcanic ash assimilation approach can be improved. This topic by itself is very technical, and arguably does not belong in the "geoscientific" journal GMD. The results may justify publication if the approach would be applicable to a large class of assimilation problems, but this is not the case as mentioned by the authors themselves. Using masks is

only applicable for a very limited set of problems, typically single point source releases of short-lived species. The problem is very idealised, with only a few observations (m=2). In this case the whole inversion problem in "observation space" is computationally very fast, which normally is not the case. The most costly step identified by the authors is a simple matrix multiplication, Aˆa = Aˆf X. Such a simple step should be very easily distributed over the available CPU's and should be very fast, so in fact I do not even understand the "problem" that the authors want to solve.

―――――――――――――――――――――

---

## Editor Comment (EC1) · R. Sander (Editor) · 23 Nov 2016

I have received two very different reviews for this manuscript: One referee only wants minor revisions, and one referee recommends rejection. Before making a final decision, I will try to find a 3rd reviewer. I would like to encourage the authors to upload a reply to the 2nd review (which may also be useful for the 3rd referee). However, I do not recommend submission of a revised manuscript until I have received the 3rd review.

---

## Author Comment (AC3) · 13 Dec 2016

Dear Anonymous Referee #2,

On behalf of all co-authors, first of all I would like to thank you for your efforts on reviewing our manuscript.

In the following we will give our response to the comments. To make the changes easier to identify, we have numbered them.

Best regards,
Guangliang Fu
on behalf all co-authors

(The revised manuscript is in the latter part of this pdf.)

**Reply to comments**:

1. *The most costly step identified by the authors is a simple matrix multiplication,*
   $A^a = A^f X$. *Such a simple step should be very easily distributed over the available CPU's and should be very fast, so in fact I do not even understand the "problem" that the authors want to solve.*

   Response:
   The problem is "we aim to accelerate the runtime time to less than the simulation time", which is important for the assimilation in an operational sense. It is described in line(s) 5.18–5.21:
   " The evaluation result of the conventional EnKF is shown in Table 1 (the middle column). It can be seen that the total computational time (4.36 h) is relatively large compared to the simulation window (3.0 h, i.e., from 9:00–12:00 UTC, 18 May, 2010), which is too much in an operational sense. Therefore, in this study, we aim to accelerate the computation to within an acceptable runtime (i.e., requires less runtime than the time period of the data assimilation application). "

   We agree that the time-consuming part $\mathbf{A^a} = \mathbf{A^f X}$ is not a complicated matrix multiplication. However, we have employed the fastest available CPU on Cartesius (Netherlands' supercomputer. Each node is configured with $2 \times$ 12-core 2.6 GHz Intel Xeon E5-2690 v3 (Haswell) CPUs and with memory 64 GB), the total computational time (4.36 h) is still not acceptable, compared to a simulation time of 3 h. This is mainly because $\mathbf{A^a} = \mathbf{A^f X}$ takes large amounts of time for the analysis step of 3.14 h. Thus, that motivated us to develop the Mask-State algorithm and the result shows it is efficient for our case. Therefore, our problem can be briefly understood as "a further acceleration of $\mathbf{A^a} = \mathbf{A^f X}$" based on model dynamics.

   In order to speed up the computation, we can do two things: (1) reduce the amount of operations in $\mathbf{A^f X}$; (2) using parallel computation. For the application at hand, the Mask-state algorithm strongly reduces the total amount of computation; parallel processing can be applied after that.

2. *This topic by itself is very technical, and arguably does not belong in the "geoscientific" journal GMD. The results may justify publication if the approach would be applicable to a large class of assimilation problems, but this is not the case as mentioned by the authors themselves. Using masks is only applicable for a very limited set of problems, typically single point source releases of short-lived species.*

Response:

Using Mask-State (MS) is applicable for many problems, where the domain is not fully polluted by the species. It is certainly not limited to "typically single point source releases of short-lived species". It doesn't matter what the emission looks like and whether the releases are "short-lived species" or "long-lived species". Given an assimilation problem, the only restriction for MS to gain an acceleration is whether the whole domain is fully polluted or partly polluted.

MS is therefore suitable for many assimilation problems, such as all the volcanic-related ash/gas assimilation, e.g., assimilation of satellite data/LIDAR data/in situ data; (sand/desert) dust storm related assimilation; tornado-related assimilation; assimilation of exploding nuclear plants or factories; chemicals or oils leaking on seas; global (forecast) fire assimilation; assimilation of environmental pollutant transport, e.g., severe smog.

For all the above applications, MS can accelerate $\mathbf{A^a} = \mathbf{A^f X}$. As a clarification, we have clearly mentioned this in the new version in line(s) 11.15–11.22:

" Using MS is also applicable for many other assimilation problems, where the domain is not fully polluted by the species. It does not matter what the emission looks like and whether the releases are short- or long-lived species. Given an assimilation problem, the only restriction for MS to gain an acceleration is whether the whole domain is fully polluted or partly polluted. The assimilation problems where MS can can achieve the acceleration effect on the computations of $\mathbf{A^a} = \mathbf{A^f X}$ are such as all the volcanic-related ash/gas assimilation, e.g., assimilation of satellite data/LIDAR data/in situ data; (sand/desert) dust storm related assimilation; tornado-related assimilation; assimilation of exploding nuclear plants or factories; chemicals or oils leaking on seas; global (forecest) fire assimilation; assimilation of environmental pollutant transport, e.g., severe smog. "

This study is submitted as a "Development and technical paper" to GMD, focusing on the development of MS to accelerate a data assimilation application not only for the volcanic ash forecast problem which is used as a case study, but also for many other assimilation problems. Therefore, we believe it fits the scope of GMD.

3. *The problem is very idealised, with only a few observations (m=2). In this case the whole inversion problem in "observation space" is computationally very fast, which normally is not the case.*

Response:

Fu et al. (2016) described the methodology for an aircraft-based volcanic ash data assimilation problem, where (for one aircraft) at each assimilation time, two in situ measurements of $PM_{10}$ and $PM_{2.5}$ are assimilated (thus $m=2$ here). The problem should not be considered as an idealised case, but an example of stationary data assimilation (although aircraft is not exactly stationary, but the measurement number at each assimilation time is steady and stationary).

We admit for $m=2$, the whole inversion problem in "observation space" is computationally fast. This is shown in Fig. 2b, where at one analysis step, the computational cost of the inversion problem ($\mathbf{X_4} = \mathbf{X_3}^{-1}$) is $O(m^3)$. However, for stationary data assimilation problems where mostly $m \ll n$ ($n$ is the state number, in this case $n \sim 10^6$), the inversion problem is also fast compared to the part $\mathbf{A^a} = \mathbf{A^f X}$ ($O(nN^2)$, $N=100$ is the ensemble size). Thus, the case $m=2$ can be taken as an example of stationary assimilation. Actually, this type of assimilation is a common assimilation type. For example, Barbu et al. (2009) assimilated measurements of $SO_2$ and $SO_4$ from the EMEP database. The assimilation set contains 17 sites for $SO_2$ and 27 for $SO_4$, thus $m=44$ ($m \ll n$).

Recently, a large number of national weather services have implemented ceilometer networks, mainly for monitoring the dispersion of volcanic ash clouds (Wiegner et al., 2014). These data set will be (and in part are already) available in near real time and will provide information about the (horizontal and) vertical distribution (with some restrictions due to cloud cover). Thus, they could be very promising candidates for stationary data assimilation for volcanic ash plumes. Under these stationary measurement setup ($m \sim 150$), based on Fig. 2b, the inversion problem is still not expensive ($O(m^3)$) compared to $\mathbf{A^a} = \mathbf{A^f X}$ ($O(nN^2)$) at one analysis step.

Therefore, for stationary/near-stationary data assimilation, it is usually the case that the inversion problem in "observation space" is computationally fast.

We also admit the problem discussed in the study is not the case where $O(m)=O(n)$, e.g., for passive satellite data assimilation, where the inversion problem in the analysis step is the most time-consuming part ($O(m^3)=O(n^3)$), but this is not the case considered in this study for stationary data assimilation. We have added some related discussion in line(s) 13.29–13.33:

[revised manuscript text omitted]

**a**

Forecast step

$\mathbf{A^f}_{n \times N}$

Analysis step
$\mathbf{A^a}_{n \times N}$

$\mathbf{O^f}_{m \times N}$

$\mathbf{Y}_{m \times N}$

Stochastic
Observer

| | |
|---|---|
| $n$ | ash state number $(3.888 \times 10^6)$ |
| $m$ | measurement number (2) |
| $N$ | ensemble size (100) |
| $\mathbf{A^f}$ | forecasted ensemble state matrix $(n \times N)$ |
| $\mathbf{O^f}$ | measured ensemble state matrix $(m \times N)$ |
| $\mathbf{Y}$ | measurement ensemble matrix $(m \times N)$ |
| $\mathbf{A^a}$ | analyzed ensemble state matrix $(n \times N)$ |
| $\mathbf{B}_{N \times N}$ | represents $\mathbf{I}_{N \times N} - \frac{1}{N}\mathbf{1}_{N \times N}$ |

**b**

**Computational cost of analysis step**

| Procedures | Cost |
|---|---|
| $\mathbf{X_1} = \mathbf{O}^f \mathbf{B}$ | $O(mN^2)$ |
| $\mathbf{X_2} = \mathbf{YB}$ | $O(mN^2)$ |
| $\mathbf{X_3} = \mathbf{X_1}\mathbf{X_1}' + \mathbf{X_2}\mathbf{X_2}'$ | $O(m^2N)$ |
| $\mathbf{X_4} = \mathbf{X_3}^{-1}$ (Singular Value Decomposition (SVD)) | $O(m^3)$ |
| $\mathbf{X_5} = \mathbf{BX_1}'$ | $O(mN^2)$ |
| $\mathbf{X_6} = \mathbf{X_5}\mathbf{X_4}$ | $O(m^2N)$ |
| $\mathbf{X} = \mathbf{I} + \mathbf{X_6}(\mathbf{Y} - \mathbf{O}^f)$ | $O(mN^2)$ |
| $\mathbf{A^a} = \mathbf{A^f}\mathbf{X}$ | $O(nN^2)$ |

$(n=3.888 \times 10^6,\ m=2,\ N=100.)$

**Figure 2. Computational evaluation of the analysis step. a**, Illustration of the analysis step. **b**, Computational cost of all sub-part of the analysis step.

[Figure]

**Figure 3. Characteristic of volcanic ash state.**

**Table 1.** Comparison of the computational cost of conventional EnKF and MS-EnKF. ( The results are obtained from the bullx B720 thin nodes of the Cartesius cluster, which is a computing facility of SURFsara, the Netherlands Supercomputing Centre. Each node is configured with $2 \times$ 12-core 2.6 GHz Intel Xeon E5-2690 v3 (Haswell) CPUs and with memory 64 GB.)

| Case | Conventional EnKF | MS-EnKF |
|------|-------------------|---------|
| Cores used | 102 | 102 |
| Tracer number ($n_{spec}$) | 6 | 6 |
| Measurements of tracers (m) | 2 | 2 |
| Ensemble size (N) | 100 | 100 |
| Parallel in forecast step | Yes | Yes |
| Parallel in analysis step | No | No |
| Mask-state in analysis step | No | Yes |
| Initialization | 0.42 h | 0.42 h |
| Forecast | 0.65 h | 0.65 h |
| Analysis | 3.14 h | 0.88 h |
| Others | 0.15 h | 0.12 h |
| Total Runtime | 4.36 h | 1.95 h |

h=hour, simulation window = 3.0 h, the time is Wall clock time.

---

## Referee Comment (RC3) · Anonymous Referee #3 · 28 Dec 2016

The authors presented a "mask-state algorithm[2] which reduces the dimension of full ensemble state matrix into a relatively smaller one and consequently reduce the computational cost in the analysis step of Ensemble Kalman Filter (EnKF) for data assimilation. Computational cost for the analysis step is studied in detail. Numerical tests show that computational times for analysis step is reduced to less than 1/3 of the original time for analysis step. The overall computational time is reduced to a level that is smaller than given time window. The idea of reducing work by only working with the non-zero rows identified by the [3]mas-state[2] is rather straightforward but the notion of exploiting sparsity and parallelism of the matrix is necessary. I am appreciative of the hard work in the complicated task of managing the irregular sparsity. Effectively this is a good problem specific reordering scheme for the matrix. This paper makes the case for the need of sparsity aware processing by comparing the performance (computation

and memory) to a full matric implementation. The overall contribution while not exciting or deep likely has an impact on data assimilation(DA) and given the standard practice in the DA community it potentially represents an advance. A true test of the quality of this work would be comparison to standard sparse matrix methods (like Compressed Sparse Row or Column or more many more advanced variants tune to multi-core and multi-processor architectures) that have been much studied (see for e.g.[1-6]) and are literally graduate textbook material.

More numerically sophisticated areas like the modeling community will find this standard but it is likely to have an impact on the DA community so I recommend publication after the authors carefully make the case for the Mask State scheme relative to more standard sparse matrix methods not just comparison to full storage dense matrices. If the authors were to attack or even attempt to tackle the harder problem of programming for parallelism with changing sparsity this would be also be a much stronger and impactful paper.

Other limitations of the proposed algorithm:

1) The proposed method is based on the fact that volcanic ash only occupies portion of the whole domain and hence there will be many zero rows in the ensemble state matrix. So such a method can only be implemented for flows that have such feature. It seems that few applications of Ensemble Kalman Filter (EnKF) data assimilation method have this feature.

2) According to [3]the analysis step takes 72% of total runtime[2], it means the analysis step is the bottle neck. However, such case might not be general for all ash forecasts. As the computational cost for initialization and forecast greatly depends on the [3]forecast model[2] that is used. If a more expensive ash forecasting model is used, then, I would guess, the bottle neck would be ash forecasting.3) It is not clear to me that a good parallel linear algebra library like PeTSC which allows users to specify their orderings and comes with machine optimized parallel matrix-matrix multiply operations
will not outperform the versions coded up here.

Modification suggestions and questions:

1) The paper focusses on "Ensemble Kalman Filter (EnKF) data assimilation method[2] and the new algorithm proposed in this paper is specific for "Ensemble Kalman Filter (EnKF) data assimilation method[2]. It would be more precise to use [3]data assimilation based on Ensemble Kalman Filter (EnKF)[2] instead of just [3]data assimilation[2] in the title.

2) What is the subscript [3]$q$[2] in Eq. (3), should it be [3]$N$[2]?

3) It would be interest to see the additional cost for reducing the size of the ensemble state matrix.

4) The "non-zero rows[2] is around 0.393 of total rows. According to cost analysis, the cost is O(nN^2). So theoretically speaking, the minimum time would be 0.393*3.14h $\sim$ 1.234h. Numerical test shows that the time goes down to 0.88h. It is interesting to know what are the other contributions to this time decrease. My guess is memory access cost also goes down when the size of matrix reduced.

5) Again, regarding questions in 4) 6) The most expensive sub-step in the analysis step is actually a matrix multiplication. It should be trivial to parallelize it by yourself or utilize libraries like scaLAPACK. This would be a more general and more powerful way to reduce time on analysis step.

7) Mathematics symbols need to be a little bit more clear, for example, in the paper, y is used as both observational space and a vector in the space?

[1] Bank, Randolph E.; Douglas, Craig C. (1993), "Sparse Matrix Multiplication Package (SMMP)"   <http://www.cs.yale.edu/homes/douglas-craig/Preprints/pub34.pdf>(PDF), Advances in Computational Mathematics, 1

[2] Buluç, Aydžn; Fineman, Jeremy T.; Frigo, Matteo; Gilbert, John R.; Leis-
erson, Charles E. <https://en.wikipedia.org/wiki/Charles_E._Leiserson> (2009). Parallel sparse matrix-vector and matrix-transpose-vector multiplication us­ing compressed sparse blocks <http://gauss.cs.ucsb.edu/~aydin/csb2009.pdf> (PDF). ACM Symp. on Parallelism in Algorithms and Architec­tures. CiteSeerX <https://en.wikipedia.org/wiki/CiteSeerX> 10.1.1.211.5256 <https://citeseerx.ist.psu.edu/viewdoc/summary?doi=10.1.1.211.5256>

[3] M. Kreutzer, G. Hager, G. Wellein, H. Fehske, and A. R. Bishop: A unified sparse matrix data format for modern processors with wide SIMD units, 2014 [4] Barrett, et al., [3]Templates for the solution of linear systems: Building Blocks for Iterative Methods, 2nd Edition[2], SIAM, 1994 (book online)

[5] Sparse BLAS standard: http://www.netlib.org/blas/blast-forum

[6] BeBOP: http://bebop.cs.berkeley.edu/
* * *

---

## Author Response (AR1)

Dear Editor,

Herewith we submit the revised manuscript gmd-2016-208, " Accelerating volcanic ash data assimilation using a mask-state algorithm based on ensemble Kalman filter: a case study with the LOTOS-EUROS model (version 1.10) ".

First we would like to thank you and three anonymous reviewers, and really appreciate the detailed comments and suggestions. We have carefully considered the comments and made changes accordingly in the revised paper. We believe the new version has been improved a lot compared to the previous version. In the new version, with the agreement of all authors, we have changed the author's order to Guangliang Fu, Hai Xiang Lin, Arnold Heemink, Sha Lu, Arjo Segers, Nils van Velzen, Tongchao Lu, Shiming Xu.

In the following we will give our answers and reactions to the comments. To make the changes easier to identify, we have numbered them.

Best regards,
Guangliang Fu
on behalf all co-authors

(The revised manuscript is in the latter part of this pdf.
Changes for Reviewer#1: "purple"; Reviewer#2: "red"; Reviewer#3: "blue".)

**Reply to Reviewer #1:**

Reply to **Specific comments**:

1. *The variable notation $N_{ash}$ should be changed to $n_{ash}$ as the small n presents number of grid points (line 4.30).*

   Response:

   I agree. All $N_{ash}$ have been changed to $n_{ash}$ in the revision.

2. *Equation (15) indicates whether we would have speedup by the MS method (line 4.5). Could it be extended to explicitly specify the expected amount of speed up (with some assumptions if necessary)?*

   Response:

   Thanks for the comment. As suggested, it is extended in **line(s) 9.19–9.23**:

   The relation $n_{ash} < \frac{N-1}{N} n$ indicates whether we would have speedup by the MS method, actually it can be extended to Eq. (15),

   $$S_{\mathrm{ms}} = \frac{\mathrm{O}(n\,N^2)}{\mathrm{O}(n\,N + n_{ash}\,N^2)} = \mathrm{O}(\frac{n}{n_{ash}}), \tag{15}$$

   which explicitly specifies the expected amount of speedup ($S_{\mathrm{ms}}$) of $\mathbf{A}^a = \mathbf{A}^f\,\mathbf{X}$ by the MS algorithm. In this case study, $N$ is taken at 100 and $\frac{n_{ash}}{n} \approx \frac{1}{3}$, so $S_{\mathrm{ms}}$ is approximately 3.0.

3. *Some explanations are necessary about the involved operations in the others part for the clarification and justification of the less (by 0.03h) computational time (Table-1).*

   Response:

   I agree. The explanations are added in **line(s) 10.19–10.23**:

   Note that it can also be observed that the computational time for the "other" parts in Table 1 (such as operations for setting environmental variables, starting and finalizing DA algorithms, as mentioned in Sect. 3.1) is slightly reduced by the MS method (i.e., 0.03 h in this case). This is because in the conventional EnKF, the ensemble mean state $\bar{\mathbf{A}}^f$ is calculated in the "other" parts as an output to finalize the DA algorithms, while in MS-EnKF, the calculations of $\bar{\mathbf{A}}^f$ are needed and directly involved in the "Analysis" part.

4. *The computational overhead involved to transform the full matrix to a small one and vice versa should be explicitly discussed. The effect in the computation time should be also discussed.*

   Response:

   As suggested, the computational overhead involved to transform the full matrix to a small one and vice versa are explicitly discussed in **line(s) 8.12–8.14**:

   **Construct masked ensemble state matrix $\tilde{\mathbf{A}}^f$**: Using the mask array $\mathbf{z}_{n_{ash} \times 1}$ obtained from step (ii), $\tilde{\mathbf{A}}^f_{n_{ash} \times N}$ can be constructed column by column according to Eq. (13), and the computational cost (overhead) for this step is $\mathrm{O}(n_{ash}\,N)$.

   $$\tilde{\mathbf{A}}^f(1:n_{ash}, 1:N) = \mathbf{A}^{\mathbf{f}}(\mathbf{z}(1:n_{ash}), 1:N), \tag{13}$$

and in **line(s) 8.17–8.20**:

**Construct analyzed ensemble state matrix $\mathbf{A}^a$**: With the computed $\tilde{\mathbf{A}}^a$ from step (iv) and the mask array $\mathbf{z}$ from step (ii), the final analyzed ensemble state matrix $\mathbf{A}^a_{n \times N}$ can be constructed based on Eq. (14). The computational cost (overhead) for this step is $\mathrm{O}(n\,N)$.

$$\mathbf{A}^a(\mathbf{z}(1:n_{ash}), 1:N) = \tilde{\mathbf{A}}^{\mathbf{a}}(1:n_{ash}, 1:N), \tag{14}$$

The effect in the computation time is discussed in **line(s) 9.5–9.10**:

Now we analyze MS's computational cost, which can be summed as $\mathrm{O}(n\,N) + \mathrm{O}(n) + \mathrm{O}(n_{ash}\,N) + \mathrm{O}(n_{ash}\,N^2) + \mathrm{O}(n\,N)$, i.e., $\mathrm{O}(n\,N + n_{ash}\,N^2)$. Thus, the computational overhead involved to transform the full matrix to a small one (i.e., $\mathrm{O}(n_{ash}\,N)$ for procedure (iii)) has little effect in the total computation cost of MS (i.e., $\mathrm{O}(n\,N + n_{ash}\,N^2)$). However, the computational overhead of transforming the small matrix to the full one (i.e., $\mathrm{O}(n\,N)$ for procedure (v)) does contribute a part, which cannot be ignored, to the total MS's computational cost.

5. *Line 3.20 : It is mentioned that 72% of the total runtime is spent on the analysis step. This information can be further extended (preferably with mathematical formulation) to predict the maximum computational speedup (e.g. 3.57 for the case in this paper). c.f. Amdahl's law (Amdahl, Gene M. (1967). "Validity of the Single Processor Approach to Achieving Large-Scale Computing Capabilities". AFIPS Conference Proceedings (30): 483-485.)*

6. *Related to the above point, a further discussion about the discrepancy in the actual speedup (e.g. 2.24 for the case in this paper) from the predicted speedup could be included.*

Response:

I agree. An extended discussion is added in **line(s) 9.24–9.32**:

According to Amdahl's law (Amdahl, 1967), the total computational speedup ($S_{\text{total}}$) by MS can be predicted by Eq. (16),

$$S_{\text{total}} = \frac{1}{(1 - p_{\text{ms}}) + \frac{p_{\text{ms}}}{S_{\text{ms}}}}, \tag{16}$$

where $p_{\text{ms}}$ is the proportion of the computational cost of $\mathbf{A}^a = \mathbf{A}^f\,\mathbf{X}$ in the overall DA computations. It has been evaluated that the computational cost of $\mathbf{A}^a = \mathbf{A}^f\,\mathbf{X}$ dominates the analysis step (see Fig. 2(b)), thus the proportion of the computational cost of $\mathbf{A}^a = \mathbf{A}^f\,\mathbf{X}$ approximates the proportion of the analysis step in the total DA computations (i.e., $p_{\text{ms}} \approx 72\,\%$ in this case, as described in Sect. 3.1). Therefore, based on Eq. (16), the maximum ("ideal") computational speedup can be predicted to be $\frac{1}{1 - p_{\text{ms}}}$ (i.e., $\approx 3.57$ for this case study) when $S_{\text{ms}}$ approximates infinity. However, this is not the actual speedup because $S_{\text{ms}}$ is in fact specified by Eq. (15). Based on discussions above, $S_{\text{total}}$ can be therefore estimated by Eq. (15) at $\approx 2.0$ in this case.

7. *It is understood from equation (15) the total speedup is quite low, is there any statistical data regarding the value of $n_{ash}/n$ to know the probability of the applicability of the proposed method and justify the weight of the work?*

Response:

Thanks for the suggestion. The related discussion is added in **line(s) 14.17–14.23**:

As stated in Eq. (15), the speedup of the MS method is approximately the inverse of $\frac{n_{ash}}{n}$. So far there is no statistical data on the value of $\frac{n_{ash}}{n}$. Considering the problem of volcanic ash transport, there is one emission point (at the volcano), all the ashes in atmospheres are transported by the directional wind drive from the same source point. Thus volcanic ash cloud is actually transported in a shape of a plume, which in general doesn't cover the full but only a small part of the 3-D domain. At the start phase of a volcanic ash eruption, $\frac{n_{ash}}{n}$ is much smaller than 1.0 (started from 0). During transport over a long time (one and a half months for this case study), $\frac{n_{ash}}{n}$ increases to approximately $\frac{1}{3}$. Therefore, the speedup of MS on volcanic ash DA will be significant.

8. *The paper investigates actually a single method (MS) to accelerate volcanic ash data assimilation. Other methods are only referred (e.g. parallelization of the analysis step) or contrasted (e.g. localization) and are not really investigated in this paper. So, the plural form (e.g. strategies) should be omitted.*

Response:

I agree. All "strategies" in the plural form have been changed to "a strategy" in the revision.

**Reply to Reviewer #2:**

Reply to comments:

1. *The most costly step identified by the authors is a simple matrix multiplication,*
   $\mathbf{A}^a = \mathbf{A}^f \mathbf{X}$. *Such a simple step should be very easily distributed over the available CPU's and should be very fast, so in fact I do not even understand the "problem" that the authors want to solve.*

   Response:
   The problem is "we aim to accelerate the runtime time to less than the simulation time", which is important for the assimilation in an operational sense. It is described in **line(s) 5.16–5.19**:

   The evaluation result of the conventional EnKF is shown in Table 1 (the middle column). It can be seen that the total computational time (4.36 h) is relatively large compared to the simulation window (3.0 h, i.e., from 9:00 - 12:00 UTC, 18 May, 2010), which is too much in an operational sense. Therefore, in this study, we aim to accelerate the computation to within an acceptable runtime (i.e., requires less runtime than the time period of the DA application).

   We agree that the time-consuming part $\mathbf{A^a} = \mathbf{A^f X}$ is not a complicated matrix multiplication. However, we have employed the fastest available CPU on Cartesius (Netherlands' supercomputer. Each node is configured with 2 × 12-core 2.6 GHz Intel Xeon E5-2690 v3 (Haswell) CPUs and with memory 64 GB), the total computational time (4.36 h) is still not acceptable, compared to a simulation time of 3 h. This is mainly because $\mathbf{A}^a = \mathbf{A}^f \mathbf{X}$ takes large amounts of time for the analysis step of 3.14 h. Thus, that motivated us to develop the Mask-State algorithm and the result shows it is efficient for our case. Therefore, our problem can be briefly understood as "a further acceleration of $\mathbf{A^a} = \mathbf{A^f X}$" based on model dynamics.

   In order to speed up the computation, we can do two things: (1) reduce the amount of operations in $\mathbf{A}^f \mathbf{X}$; (2) using parallel computation. For the application at hand, the Mask-state algorithm strongly reduces the total amount of computation; parallel processing can be applied after that.

2. *This topic by itself is very technical, and arguably does not belong in the "geoscientific" journal GMD. The results may justify publication if the approach would be applicable to a large class of assimilation problems, but this is not the case as mentioned by the authors themselves. Using masks is only applicable for a very limited set of problems, typically single point source releases of short-lived species.*

   Response:
   Using Mask-State (MS) is applicable for many problems, where the domain is not fully polluted by the species. It is certainly not limited to "typically single point source releases of short-lived species". It doesn't matter what the emission looks like and whether the releases are "short-lived species" or "long-lived species". Given an assimilation problem, the only restriction for MS to gain an acceleration is whether the whole domain is fully polluted or partly polluted.

   MS is therefore suitable for many assimilation problems, such as all the volcanic-related ash/gas assimilation, e.g., assimilation of satellite data/LIDAR data/in situ data; (sand/desert) dust storm related assimilation; tornado-related assimilation; assimilation of exploding nuclear plants or factories; chemicals or oils leaking on seas; global (forecast) fire assimilation; assimilation of environmental pollutant transport, e.g., severe smog. In addition, for data assimilation applications (e.g., Ozone, $SO_2$) where pollutants spread over the whole domain, usually the focus is only on the high concentrations and a threshold can be set to ignore the very low values without losing the necessary

assimilation accuracy. In this case, MS can also lead to a potential acceleration since many very low concentrations can be explicitly truncated to be zeros.

For all the above applications, MS can accelerate $\mathbf{A}^a = \mathbf{A}^f \mathbf{X}$. As a clarification, we have clearly mentioned this in the new version in **line(s) 13.26–14.3**:

Using MS is also applicable for many other DA problems, where the domain is not fully polluted by the species. It does not matter what the emission looks like and whether the releases are short- or long-lived species. Given an assimilation problem, the only restriction for MS to gain an acceleration is whether the whole domain is fully polluted or partly polluted. The assimilation problems where MS can achieve the acceleration effect on the computations of $\mathbf{A}^a = \mathbf{A}^f \mathbf{X}$ include all the volcanic-related ash/gas assimilation, e.g., assimilation of satellite data/LIDAR data/in situ data; (sand/desert) dust storm related assimilation; tornado-related assimilation; assimilation of exploding nuclear plants or factories; chemicals or oils leaking on seas; global (forecast) fire assimilation; assimilation of environmental pollutant transport, e.g., severe smog. In addition, for DA applications (e.g., Ozone, $SO_2$) where pollutants spread over the whole domain, usually the focus is only on the high concentrations and a threshold can be set to ignore the very low values without losing the necessary assimilation accuracy. In this case, MS can also lead to a potential acceleration since many very low concentrations can be explicitly truncated to be zeros.

This study is submited as a "Development and technical paper" to GMD, focusing on the development of MS to accelerate a data assimilation application not only for the volcanic ash forecast problem which is used as a case study, but also for many other assimilation problems. Therefore, we believe it fits the scope of GMD.

3. *The problem is very idealised, with only a few observations (m=2). In this case the whole inversion problem in "observation space" is computationally very fast, which normally is not the case.*

Response:
Fu et al. (2016) described the methodology for an aircraft-based volcanic ash data assimilation problem, where (for one aircraft) at each assimilation time, two in situ measurements of $PM_{10}$ and $PM_{2.5}$ are assimilated (thus $m=2$ here). The problem should not be considered as an idealised case, but an example of stationary data assimilation (although aircraft is not exactly stationary, but the measurement number at each assimilation time is steady and stationary).

We admit for $m=2$, the whole inversion problem in "observation space" is computationally fast. This is shown in Fig. 2(b), where at one analysis step, the computational cost of the inversion problem ($\mathbf{X_4} = \mathbf{X_3}^{-1}$) is $O(m^3)$. However, for stationary data assimilation problems where mostly $m \ll n$ ($n$ is the state number, in this case $n \sim 10^6$), the inversion problem is also fast compared to the part $\mathbf{A}^a = \mathbf{A}^f \mathbf{X}$ ($O(n\,N^2)$, $N=100$ is the ensemble size). Thus, the case $m=2$ can be taken as an example of stationary assimilation. Actually, this type of assimilation is a common assimilation type. For example, Barbu et al. (2009) assimilated measurements of $SO_2$ and $SO_4$ from the EMEP database. The assimilation set contains 17 sites for $SO_2$ and 27 for $SO_4$, thus $m=44$ ($m \ll n$).

Recently, a large number of national weather services have implemented ceilometer networks, mainly for monitoring the dispersion of volcanic ash clouds (Wiegner et al., 2014). These data set will be (and in part are already) available in near real time and will provide information about the (horizontal and) vertical distribution (with some restrictions due to cloud cover). Thus, they could be very promising candidates for stationary data assimilation for volcanic ash plumes. Under these stationary measurement setup ($m \sim 150$), based on Fig. 2(b), the inversion problem is still not expensive ($O(m^3)$) compared to $\mathbf{A}^a = \mathbf{A}^f \mathbf{X}$ ($O(n\,N^2)$) at one analysis step.

Therefore, for stationary/near-stationary data assimilation, it is usually the case that the inversion problem in "observation space" is computationally fast.

We also admit the problem discussed in the study is not the case where $O(m)=O(n)$, e.g., for passive satellite data assimilation, where the inversion problem in the analysis step is the most time-consuming part ($O(m^3)=O(n^3)$), but this is not the case considered in this study for stationary data assimilation. We have added some related discussion in **line(s) 17.2–17.7**:

However, in some applications when many measurements are assimilated (e.g., satellite-based or seismic-based data), and the number of measurements is of the same order as the number of state variables, the most time-consuming part will be the SVD. In these cases, the contributions of MS will be limited. The reduction of the total computing time using MS therefore is less significant, an effective acceleration algorithm for the analysis step must be used and should consider the computationally-expensive SVD in the first place.

**Reply to Reviewer #3:**

Reply to **general comments**:

1. *More numerically sophisticated areas like the modeling community will find this standard but it is likely to have an impact on the DA community so I recommend publication after the authors carefully make the case for the Mask State scheme relative to more standard sparse matrix methods not just comparison to full storage dense matrices.*

    Response:

    We agree that the previous only comparison of the mask-state algorithm (MS) to the cases of full storage dense matrices was not enough. The comparison of MS to more standard sparse matrix methods (e.g., CSR-based sparse-dense matrix multiplication (SDMM)) is necessary, because MS can also be taken as one specific sparse matrix method, which typically works for ensemble-based data assimilation (DA) applications.

    We have included a comparison in the revision where the case of MS is compared with standard sparse matrix methods. The detailed comparison and explanations are presented in the new Sect. 5 (two new tables (Table 3 to 4) are added to show the result), in **line(s) 11.16–13.22**, as shown in the following.

[revised manuscript text omitted]

2. *If the authors were to attack or even attempt to tackle the harder problem of programming for parallelism with changing sparsity this would be also be a much stronger and impactful paper.*

Response:

In the revision, we have added Sect. 6.3 to discuss on the parallelization issues. In this paper, for the current usage, we keep the possibility of parallelization open, becuase a serial MS has been shown to be efficient already. Therefore, although (from application's aspect) further acceleration by parallelization is not urgent, we fully agree that a detailed discussion on parallelization in MS or CSR-based SDMM is useful and will have a good added value to the presentation of the serial MS. The discussion is in **line(s) 15.19–16.8** in the paper as follows,

**6.3 MS and parallelization**

Motivated from the model's physics, the implementation MS currently is for the serial case. This implementation has reduced the computation time to an acceptable time (i.e., simulation time is less than the period of forecast in real world time). It is however interesting to discuss the potential of parallelization of the dense-dense matrix multiplication ($\tilde{\mathbf{A}}^a_{n_{ash} \times N} = \tilde{\mathbf{A}}^f_{n_{ash} \times N} \mathbf{X}_{N \times N}$) in the step (iv) of the algorithm (see Sect. 4.2 and Table 2). The related matrix multiplication can be easily parallelized on multiple processors. Optimization and evaluation on the parallelized MS will be considered in future. For the current case study, the computational time (3.13 h, see Table 3) for an "ideal" reduction by parallelization of MS is not much larger than the acceleration (already) gained by MS (2.26 h, subtraction between 3.13 h and 0.87 h, see Table 3). Therefore, from application's perspective, further acceleration by parallelization is not required.

Alternatively, one may also consider to (1) directly parallelize the expensive matrix multiplication of $\mathbf{A}^a_{n \times N} = \mathbf{A}^f_{n \times N} \mathbf{X}_{N \times N}$, without first performing MS; or (2) implement CSR-based SDMM (see Sect. 5) with parallelization. Both are possible alternative approaches to accelerate the expensive matrix multiplication. The first approach can be implemented by user own designed parallelization, or by utilizing scaLAPACK (`https://www.netlib.org/scalapack/`, where the main function is "pdgemm"). The second approach can be realized by using some general parallel sparse-dense matrix multiplication methods (e.g., sending each column of X and three CSR arrays of $\mathbf{A}^f$ to each processor to calculate each column of $\mathbf{A}^a$) or using a good parallel algebra library like PeTSC (`https://www.mcs.anl.gov/petsc/`) which allows users to specify own orderings and comes with machine optimized parallel matrix-matrix multiplication operations. However, given the fact that MS can also be parallelized using similar ways or same libraries, thus it is fair to not consider parallelization for all cases (i.e., using MS, not using MS, using CSR-based SDMM). Actually, the parallelization in MS could be performed much easier than other approaches in dealing with $\mathbf{A}^a = \mathbf{A}^f \mathbf{X}$, because the dense-dense matrix multiplication (parallelization in the step (iv) of MS) is easier to parallelize than the sparse-dense matrix multiplication (direct parallelization for

$\mathbf{A}^a = \mathbf{A}^f\,\mathbf{X}$ or parallelized CSR-based SDMM).

In this paper, for the current usage, we keep the possibility of parallelization open, because a serial MS has been efficient already.

In addition, in the ash forecast step, due to the changing sparsity in $\mathbf{A}^f$, a fixed domain-decomposed parallelization to compute $\mathbf{A}^f$ usually would not work efficiently. In the future if further parallelization is required to deal with large domain and more complicated forecast step, some advanced approach such as adaptive domain-decomposed parallelization (Lin et al., 1998) might be employed. We have also discussed on this issue in **line(s) 11.9–11.15**:

As for domain-decomposed parallelization (Segers, 2002), it is not efficient for our application. This is because volcanic ash is special in the sense that the model is only doing computations in a small part of the domain (i.e., there is no data in a rather large part of domain), and this active part is continuously changing. Thus, a fixed domain decomposition is not very useful here because of the changing plume position. In this sense, some advanced approach such as adaptive domain-decomposed parallelization (Lin et al., 1998) should be adopted to achieve additional acceleration to the volcanic ash forecast stage. This is an interesting subject for future application, when a more complicated model is employed, only ensemble parallelization may be not enough for the forecast stage.

**Reply to **other limitations of the proposed algorithm**:**

1. *The proposed method is based on the fact that volcanic ash only occupies portion of the whole domain and hence there will be many zero rows in the ensemble state matrix. So such a method can only be implemented for flows that have such feature. It seems that few applications of Ensemble Kalman Filter (EnKF) data assimilation method have this feature.*

   Response:

   Thanks for the comment. We agree that many EnKF data assimilation applications don't have this feature. However, there are many other DA applications featuring as volcanic ash DA. We have mentioned these applications in the revision, as in **line(s) 13.29–14.3**:

   The assimilation problems where MS can achieve the acceleration effect on the computations of $\mathbf{A}^a = \mathbf{A}^f\,\mathbf{X}$ include all the volcanic-related ash/gas assimilation, e.g., assimilation of satellite data/LIDAR data/in situ data; (sand/desert) dust storm related assimilation; tornado-related assimilation; assimilation of exploding nuclear plants or factories; chemicals or oils leaking on seas; global (forecast) fire assimilation; assimilation of environmental pollutant transport, e.g., severe smog. In addition, for DA applications (e.g., Ozone, $SO_2$) where pollutants spread over the whole domain, usually the focus is only on the high concentrations and a threshold can be set to ignore the very low values without losing the necessary assimilation accuracy. In this case, MS can also lead to a potential acceleration since many very low concentrations can be explicitly truncated to be zeros.

2. *According to "the analysis step takes 72% of total runtime", it means the analysis step is the bottle neck. However, such case might not be general for all ash forecasts. As the computational cost for initialization and forecast greatly depends on the "forecast model" that is used. If a more expensive ash forecasting model is used, then, I would guess, the bottle neck would be ash forecasting.*

   Response:

   Agree. We have discussed on this point in the revision, as in **line(s) 16.25–16.33**:

   In this case study with the LOTOS-EUROS model (version 1.10), after the parallelization is

performed for the forecast step of EnKF assimilation, the analysis step takes 72 % of total run-time, which means the analysis step is the bottle neck. This case might not be general for all ash forecasts. As the computational cost for initialization and forecast greatly depends on the forecast model that is used. For the current development, it makes sense to use the LOTOS-EUROS model, because the model has been configured and evaluated in (Fu et al., 2015) by comparison with other famous models (e.g., NAME (Jones et al., 2007), WRF-Chem (Webley et al., 2012)) in simulating volcanic ash transport. However, if a more expensive ash forecasting model is used, then the bottle neck would be the forecast step. In this case, the forecast step should be the goal for acceleration and probably a parallel model or adaptive domain-decomposition (as discussed in Sect. 4.3) needs to be employed together with the parallel ensemble forecasts.

3. *It is not clear to me that a good parallel linear algebra library like PeTSC which allows users to specify their orderings and comes with machine optimized parallel matrix-matrix multiply operations will not outperform the versions coded up here.*

Response:

I agree that a direct parallelization (e.g., using PeTSC) for spare-dense matrix multiplication of $\mathbf{A}^f \mathbf{X}$ can probably outperform a serial MS. However, given the fact that MS can also be parallelized using similar ways or the same libraries, thus it is fair to not consider parallelization for all cases (i.e., using MS, not using MS, using CSR-based SDMM).

Actually, the parallelization in MS could be performed much easier than other approaches in dealing with $\mathbf{A}^a = \mathbf{A}^f \mathbf{X}$, because the dense-dense matrix multiplication (parallelization in the step (iv) of MS) is usually less complicated to handle than parallelization of the sparse-dense matrix multiplication (direct parallelization for $\mathbf{A}^a = \mathbf{A}^f \mathbf{X}$ or parallelized CSR-based SDMM).

In this paper, for the current usage, we keep parallelization open (a serial MS has been efficient already), since many parallelization strategies can be performed/adjusted within MS. It is for sure that the parallelization's effect on MS will be positive. However, whether a parallel version of MS is generally required (with respect to the extra efforts in parallelizing MS) depends on assimilation problems and needs to be investigated in future.

We agree that above aspects were not clear in the previous version. We have included them in **line(s) 15.28–16.8**:

Alternatively, one may also consider to (1) directly parallelize the expensive matrix multiplication of $\mathbf{A}^a_{n \times N} = \mathbf{A}^f_{n \times N} \mathbf{X}_{N \times N}$, without first performing MS; or (2) implement CSR-based SDMM (see Sect. 5) with parallelization. Both are possible alternative approaches to accelerate the expensive matrix multiplication. The first approach can be implemented by user own designed parallelization, or by utilizing scaLAPACK (https://www.netlib.org/scalapack/, where the main function is "pdgemm"). The second approach can be realized by using some general parallel sparse-dense matrix multiplication methods (e.g., sending each column of X and three CSR arrays of $\mathbf{A}^f$ to each processor to calculate each column of $\mathbf{A}^a$) or using a good parallel algebra library like PeTSC (https://www.mcs.anl.gov/petsc/) which allows users to specify own orderings and comes with machine optimized parallel matrix-matrix multiplication operations. However, given the fact that MS can also be parallelized using similar ways or same libraries, thus it is fair to not consider parallelization for all cases (i.e., using MS, not using MS, using CSR-based SDMM). Actually, the parallelization in MS could be performed much easier than other approaches in dealing with $\mathbf{A}^a = \mathbf{A}^f \mathbf{X}$, because the dense-dense matrix multiplication (parallelization in the step (iv) of MS) is easier to parallelize than the sparse-dense matrix multiplication (direct parallelization for $\mathbf{A}^a = \mathbf{A}^f \mathbf{X}$ or parallelized CSR-based SDMM).

In this paper, for the current usage, we keep the possibility of parallelization open, because a serial MS has been efficient already.

Reply to **modification suggestions and questions**:

1. *The paper focusses on "Ensemble Kalman Filter (EnKF) data assimilation method" and the new algorithm proposed in this paper is specific for "Ensemble Kalman Filter (EnKF) data assimilation method". It would be more precise to use " data assimilation based on Ensemble Kalman Filter (EnKF)" instead of just "data assimilation" in the title.*

   Response:

   Agree. As suggested, the title has been re-written in the revision,

   Accelerating volcanic ash data assimilation using a mask-state algorithm based on ensemble Kalman filter: a case study with the LOTOS-EUROS model (version 1.10)

2. *What is the subscript "q" in Eq. (3), should it be "N" ?*

   Response:

   Yes and corrected in **line(s) 4.6**.

3. *It would be interest to see the additional cost for reducing the size of the ensemble state matrix.*

   Response:

   Thanks for the comment. In the revision, we have added Table 2 to specify the computational time for all five steps of MS. And we have also added discusion related to this point, as in **line(s) 10.10–10.17**:

   The detailed experimental time for each step of MS is shown in Table 2. As expected, the dense-dense matrix multiplication in step (iv) takes the largest part (i.e., 0.8474 h for this case study) of the total computational time (0.87 h) of MS. However, step (iv) has been a big improvement compared to the case without MS (3.14 h, see Table 1), which is because the computational time for other steps (e.g., steps (i - iii) costs only 0.0156 h to reduce the size of the ensemble state matrix) is little and ignorable. Note that the total computational time of $\mathbf{A}^a = \mathbf{A}^f \mathbf{X}$ with MS (i.e., 0.87 h in Table 2) is not exactly equal to the compuational time of the MS-EnKF analysis procedures (i.e., 0.88 h in Table 1). The subtraction (i.e., 0.01 h) corresponds to the summed computational time of all the other analysis procedures (i.e., procedures 1 - 8) except for $\mathbf{A}^a = \mathbf{A}^f \mathbf{X}$ (see Fig. 2(b) and Table 3).

4. *The "non-zero rows" is around 0.393 of total rows. According to cost analysis, the cost is $O(nN^2)$. So theoretically speaking, the minimum time would be 0.393*3.14 h ∼ 1.234 h. Numerical test shows that the time goes down to 0.88h. It is interesting to know what are the other contributions to this time decrease. My guess is memory access cost also goes down when the size of matrix reduced.*

   Response:

   This is a good point. One reason is that when the size of matrix is reduced, the memory access cost also goes down (e.g., through better cache usages). Another possible reason is that the ash grid number actually decreases with time (not always taking 0.393 the total grid number), due to ash sedimentation and deposition processes ((Fu et al., 2015)).

   We have discussed this point in **line(s) 10.33–11.4**:

There is another interesting point. According to Fig. 3, the ash grids comprises $39.3\,\%$ of the total grids. Thus, the minimum computing time by using MS to utilize this model's characteristic should be $\approx 1.234\,\mathrm{h}$ (i.e., $0.393 \times 3.14\,\mathrm{h}$). However, the experimental result shows that the computational time goes down to $0.88\,\mathrm{h}$ (see Table 1). One reason for this time decrease is that when the size of matrix is reduced, the memory access cost also goes down (e.g., through better cache usages). Another possible reason is that the ash grid number actually decreases with time (not always taking $39.3\,\%$ the total grid number), due to ash sedimentation and deposition processes ((Fu et al., 2015)).

5. *Again, regarding questions in 4,*

6. *the most expensive sub-step in the analysis step is actually a matrix multiplication. It should be trivial to parallelize it by yourself or utilize libraries like scaLAPACK. This would be a more general and more powerful way to reduce time on analysis step.*

**Response to 5 and 6:**

In order to speed up the matrix multiplication, we can do two things: (1) reduce the amount of operations in $\mathbf{A}^f\,\mathbf{X}$; (2) using parallel computation. For the application at hand, MS strongly reduces the total amount of computation; parallel processing can be applied after that.

In this study, although parallelization in MS is not necessarily required, we have carefully discussed its potential, in **line(s) 15.20–16.6**:

Motivated from the model's physics, the implementation MS currently is for the serial case. This implementation has reduced the computation time to an acceptable time (i.e., simulation time is less than the period of forecast in real world time). It is however interesting to discuss the potential of parallelization of the dense-dense matrix multiplication ($\tilde{\mathbf{A}}^a_{n_{ash} \times N} = \tilde{\mathbf{A}}^f_{n_{ash} \times N}\mathbf{X}_{N \times N}$) in the step (iv) of the algorithm (see Sect. 4.2 and Table 2). The related matrix multiplication can be easily parallelized on multiple processors. Optimization and evaluation on the parallelized MS will be considered in future. For the current case study, the computational time ($3.13\,\mathrm{h}$, see Table 3) for an "ideal" reduction by parallelization of MS is not much larger than the acceleration (already) gained by MS ($2.26\,\mathrm{h}$, subtraction between $3.13\,\mathrm{h}$ and $0.87\,\mathrm{h}$, see Table 3). Therefore, from application's perspective, further acceleration by parallelization is not required.

Alternatively, one may also consider to (1) directly parallelize the expensive matrix multiplication of $\mathbf{A}^a_{n \times N} = \mathbf{A}^f_{n \times N}\mathbf{X}_{N \times N}$, without first performing MS; or (2) implement CSR-based SDMM (see Sect. 5) with parallelization. Both are possible alternative approaches to accelerate the expensive matrix multiplication. The first approach can be implemented by user own designed parallelization, or by utilizing scaLAPACK (`https://www.netlib.org/scalapack/`, where the main function is "pdgemm"). The second approach can be realized by using some general parallel sparse-dense matrix multiplication methods (e.g., sending each column of X and three CSR arrays of $\mathbf{A}^f$ to each processor to calculate each column of $\mathbf{A}^a$) or using a good parallel algebra library like PeTSC (`https://www.mcs.anl.gov/petsc/`) which allows users to specify own orderings and comes with machine optimized parallel matrix-matrix multiplication operations. However, given the fact that MS can also be parallelized using similar ways or same libraries, thus it is fair to not consider parallelization for all cases (i.e., using MS, not using MS, using CSR-based SDMM). Actually, the parallelization in MS could be performed much easier than other approaches in dealing with $\mathbf{A}^a = \mathbf{A}^f\,\mathbf{X}$, because the dense-dense matrix multiplication (parallelization in the step (iv) of MS) is easier to parallelize than the sparse-dense matrix multiplication (direct parallelization for $\mathbf{A}^a = \mathbf{A}^f\,\mathbf{X}$ or parallelized CSR-based SDMM).

7. *Mathematics symbols need to be a little bit more clear, for example, in the paper, y is used as both observational space and a vector in the space?*

**Response:**

Agree. In the revision, we have checked all the mathematics symbols. The observation vector $\boldsymbol{y}$ is now clearly mentioned just as a vector, as in **line(s) 4.13–4.14**:

[revised manuscript text omitted]